# FedPS: Federated Preprocessing for structured data via aggregated Statistics

## Abstract

Federated Learning (FL) enables multiple parties to collaboratively train machine learning models without sharing raw data. However, before training, data must be preprocessed to address missing values, inconsistent formats, and heterogeneous feature scales. This preprocessing stage is critical for model performance but is largely overlooked in FL research. In practical FL systems, privacy constraints prohibit centralizing raw data, while communication efficiency introduces further challenges for distributed preprocessing. We introduce FedPS, a framework for federated data preprocessing based on aggregated statistics. FedPS leverages data-sketching techniques to efficiently summarize local datasets while preserving essential statistical information. Building on these summaries, we design federated algorithms for feature scaling, encoding, discretization, and missing-value imputation, and extend preprocessing-related models such as Bayesian Linear Regression to both horizontal and vertical FL settings. FedPS provides flexible, communication-efficient, and consistent preprocessing pipelines for practical FL deployments.

## 1 Introduction

Data preprocessing (García et al., 2016) is a vital stage of the machine learning pipeline, transforming raw inputs into clean, structured, and analyzable forms. In structured data, e.g., tabular data, common preprocessing tasks include handling missing values, normalizing feature scales, and encoding categorical variables. Effective preprocessing improves model accuracy, accelerates convergence, and enhances interpretability. Yet despite its importance to model performance, preprocessing remains largely neglected in federated learning (Kairouz et al., 2021), where multiple entities collaboratively train a model on decentralized data.

Most federated learning research focuses on improving training algorithms (McMahan et al., 2017; Stich, 2019; Karimireddy et al., 2020; Li et al., 2020a;b), typically assuming that data has already been cleaned and transformed. This assumption hides a significant practical bottleneck: without consistent preprocessing (meaning that all clients apply identical preprocessing parameters derived from global knowledge), even state-of-the-art federated learning algorithms fail to achieve their full potential. In practical federated systems, preprocessing introduces distinct challenges. Privacy constraints prohibit centralizing raw data for joint preparation, communication efficiency limits the information clients can exchange, and data heterogeneity across clients complicates the design of consistent preprocessing pipelines.

We discuss several possible strategies for preprocessing in FL and highlight their limitations.

**Option 1: Centralized preprocessing**. Many simulation-based FL studies preprocess the data centrally before partitioning it among clients. While this ensures consistent preprocessing and strong baselines, it is infeasible in real deployments. Centralized preprocessing requires collecting raw data, which directly violates FL's foundational privacy constraints. Thus, it is suitable only for simulation, not practical for FL.

**Option 2: No preprocessing**. One may simply train on raw, unprocessed data, avoiding privacy issues but severely compromising model performance. Real-world data is often incomplete, inconsistent, or heterogeneously scaled, all of which harm convergence and generalization. As shown in Section 5, models trained without preprocessing perform substantially worse than those using properly prepared data.

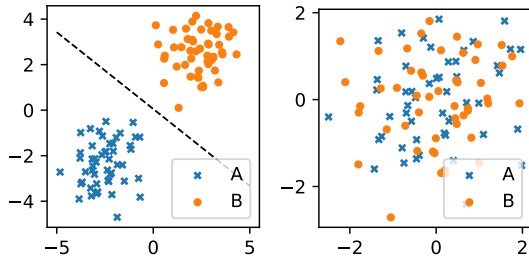

Figure 1: Clients A and B hold data with different label distributions. The raw data is linearly separable (left). After each client applies local standardization, the combined data becomes no longer linearly separable (right).

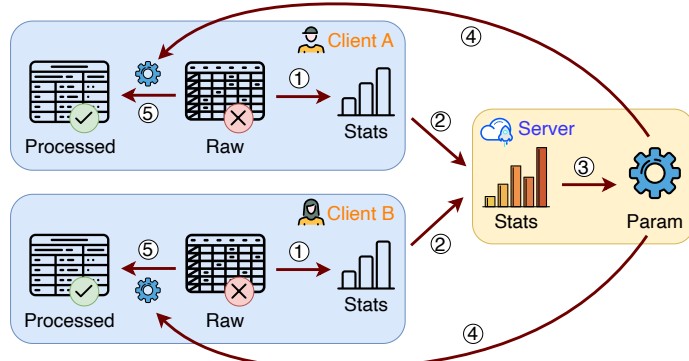

Figure 2: Overview of federated data preprocessing in FedPS.

**Option 3: Transfer preprocessing**. Another approach is to reuse preprocessing parameters derived from public datasets or pretrained models (Qi & Wang, 2024). While this avoids learning from private data, its success hinges on the similarity between public and private datasets, an unrealistic assumption in heterogeneous FL environments. Transfer techniques may handle basic format alignment but fail for distribution-dependent tasks such as imputation or discretization.

**Option 4: Local preprocessing**. Each client may preprocess its own data locally, preserving privacy but sacrificing cross-client consistency due to heterogeneous data distributions. In non-IID (independent and identically distributed) settings (Li et al., 2022), local transformations (e.g., normalization) can distort the global data distribution. Figure 1 illustrates how independent standardization can render previously well-separated classes non-separable. Our experiments (Section 5) show that inconsistent local preprocessing may degrade performance even below using raw data, underscoring the need for coordinated preprocessing.

**Option 5: Federated preprocessing**. Federated preprocessing overcomes the limitations above by coordinating preprocessing without exposing raw data. As illustrated in Figure 2, this approach consists of five steps: ① Compute local statistics; ② Share and aggregate statistics; ③ Derive preprocessing parameters; ④ Broadcast parameters to clients; ⑤ Apply preprocessing locally. This paradigm ensures consistency, keeps raw data local, and addresses challenges posed by non-IID data distributions.

We present FedPS, a federated preprocessing framework for structured data and open-source library[1] that maps the Scikit-learn preprocessing operations to the federated setting. . FedPS comprises two complementary components: (1) a general workflow for federated preprocessing based on aggregated statistics, illustrated in Figure 2, and (2) a comprehensive suite of preprocessing methods that instantiate this workflow. To demonstrate the framework's flexibility and practical utility, we implement a broad range of preprocessing techniques spanning scaling, encoding, transformation, discretization, and imputation. These methods leverage data-sketching techniques (Cormode & Yi, 2020) to support communication-efficient computation (independent of dataset size) of complex global statistics (e.g., quantiles, frequent items). We further extend core preprocessing-related models, including $k$-Means (Lloyd, 1982), $k$-Nearest Neighbors, and Bayesian Linear Regression (Tipping, 2001), to both horizontal and vertical FL settings, making the library flexible and applicable across diverse federated data preprocessing scenarios.

Existing federated learning frameworks already support a small subset of preprocessing methods, such as normalization and one-hot encoding (Liu et al., 2021; The SecretFlow Authors, 2022; Baunsgaard et al., 2021; 2022). Our goal is not to re-invent these individual algorithms. Instead, FedPS provides a framework that systematically supports a wider collection of preprocessing methods by identifying their sufficient statistics and mapping them to reusable federated computation primitives. When suitable federated algorithms already exist (e.g., federated $k$-Means and $k$-Nearest Neighbors), FedPS incorporates them. For preprocessing operations lacking existing federated formulations, such as the Bayesian linear regression used by *IterativeImputer*, we derive new federated algorithms.

---

[1] https://anonymous.4open.science/r/fedps/README.md

Our main contributions are summarized as follows:

- We introduce FedPS, a framework for federated preprocessing that maintains consistency across clients through summarization, aggregation, and parameter distribution (Section 3.2).

- We implement comprehensive preprocessing methods leveraging data-sketching techniques for communication-efficient computation of complex global statistics (Section 3.2).

- We analyze the sufficient statistics and communication costs required by different preprocessing operations, providing practical guidance for scalable deployment in federated settings (Section 3.3).

- We develop federated Bayesian linear regression for both horizontal and vertical settings, enabling sophisticated model-based preprocessing while avoiding cross-client feature interactions (Section 4).

- Our empirical results across various datasets indicate that the accuracy of federated preprocessing significantly surpasses both local-only preprocessing and raw data baselines, particularly in heterogeneous data contexts, thereby confirming the efficacy of federated preprocessing (Section 5).

The rest of the paper is organized as follows. Section 2 reviews techniques foundational to federated preprocessing. Section 3 presents the FedPS framework. Section 4 develops federated Bayesian linear regression. Section 5 reports empirical results, followed by discussion in Section 6 and conclusions in Section 7.

## 2 Preliminaries

### 2.1 Data Preprocessing

Data preprocessing refers to the collection of techniques used to prepare raw data for downstream analysis or modeling. In tabular data, each column represents a feature, and preprocessing is often applied to individual columns, although operations involving multiple columns are also possible. Common steps include feature scaling, encoding, discretization, imputation of missing values, and other transformations tailored to specific learning tasks. Mature software packages such as *Scikit-learn* (Pedregosa et al., 2011) provide standardized and reliable implementations of these techniques.

### 2.2 Federated Learning

Federated learning is a distributed paradigm where multiple clients collaboratively train a model without sharing their raw data. A central challenge is communication cost, since exchanging large volumes of information can be inefficient. The FedAvg algorithm (McMahan et al., 2017) mitigates this by allowing several rounds of local updates before aggregation, thereby reducing the required number of communication rounds. Another difficulty is data heterogeneity. Clients may hold data drawn from different distributions, and these differences can lead to divergent updates when local models are combined by simple averaging. A variety of methods have been proposed to improve stability under heterogeneous data (Li et al., 2020a;b).

Federated learning is typically divided into two settings based on data partitioning. Horizontal FL is when clients share the same feature space but hold different examples. Vertical FL is when clients share a common identifier space but possess different feature spaces. Since most preprocessing methods operate on each feature separately, our main focus is on the horizontal setting, with vertical extensions introduced when required.

### 2.3 Statistics Aggregation

In federated learning, each client computes local statistics or summaries and transmits them to the server, which then aggregates them into global quantities. Basic statistics such as the minimum, maximum, sum, mean, and variance can be efficiently computed in a distributed manner with minimal communication. Set union can also be performed by merging local sets. More complex quantities, such as quantiles and frequent items, require specialized algorithms to achieve both accuracy and communication efficiency.

**Quantiles.** Quantiles divide the dataset into equal sized partitions. Computing them exactly requires storing the entire dataset (Munro & Paterson, 1980), which is impractical in federated settings due to the associated communication cost. Approximate quantile algorithms based on data sketches are therefore preferred. Examples include the KLL sketch (Karnin et al., 2016), which provides additive error guarantees, and the REQ sketch (Cormode et al., 2023), which provides multiplicative error guarantees. Both are designed to operate efficiently in distributed environments.

**Frequent Items.** Identifying the most common items in a dataset requires counting all occurrences, which is expensive when performed exactly. Frequent item sketches (Anderson et al., 2017) provide a compact approximation. In our implementation, we use the *DataSketches* library (The DataSketches Authors, 2023), which offers an effective balance between accuracy and communication cost.

## 2.4  Bayesian Linear Regression

Bayesian linear regression (BLR) (Tipping, 2001; Bishop, 2006) is an important tool within *IterativeImputer*, used for imputation of missing values. It formulates linear regression within a probabilistic framework by placing a prior distribution over the model parameters $\boldsymbol{\omega}$. A common choice is an isotropic Gaussian prior with zero mean $p(\boldsymbol{\omega} \mid \alpha) = \mathcal{N}(\boldsymbol{\omega} \mid \mathbf{0}, \alpha^{-1}\mathbf{I})$, where $\alpha$ denotes the prior precision. Given the data matrix $\mathbf{X}$ and label $\mathbf{Y}$, and assuming Gaussian noise precision $\beta$, the posterior distribution over $\boldsymbol{\omega}$ is also Gaussian: $p(\boldsymbol{\omega} \mid \mathbf{X}, \mathbf{Y}, \beta) = \mathcal{N}(\boldsymbol{\omega} \mid \hat{\boldsymbol{\omega}}, \boldsymbol{\Sigma})$, with posterior mean and covariance given by:

$$\hat{\boldsymbol{\omega}} = \beta\boldsymbol{\Sigma}^{-1}\mathbf{X}^{\top}\mathbf{Y}, \quad \boldsymbol{\Sigma} = \alpha\mathbf{I} + \beta\mathbf{X}^{\top}\mathbf{X}. \tag{1}$$

The hyperparameters $\alpha$ and $\beta$ follow Gamma hyperpriors $p(\alpha) = \mathrm{Gamma}(\alpha \mid a_1, a_2)$, $p(\beta) = \mathrm{Gamma}(\beta \mid b_1, b_2)$, where $\mathrm{Gamma}(\alpha \mid a, b) = \Gamma(a)^{-1}b^a\alpha^{a-1}e^{-b\alpha}$. Since $\alpha$ and $\beta$ cannot be computed in closed form, they are updated iteratively together with $\hat{\boldsymbol{\omega}}$ and $\boldsymbol{\Sigma}$ as described in Appendix A of Tipping (2001):

$$\alpha = \frac{n - \gamma + 2a_1}{\varepsilon + 2a_2}, \quad \beta = \frac{\gamma + 2b_1}{\|\hat{\boldsymbol{\omega}}\|_2^2 + 2b_2}, \quad \gamma = \sum_i \frac{\alpha\boldsymbol{\Lambda}_i}{\beta + \alpha\boldsymbol{\Lambda}_i}, \quad \varepsilon = \|\mathbf{Y} - \mathbf{X}\hat{\boldsymbol{\omega}}\|_2^2. \tag{2}$$

Here, $\boldsymbol{\Lambda}_i$ denotes the $i$-th eigenvalue of $\mathbf{X}^{\top}\mathbf{X}$. Equivalently, letting $\mathbf{X} = \mathbf{U}\mathbf{S}\mathbf{V}^{\top}$ be the singular value decomposition of $\mathbf{X}$, we have $\boldsymbol{\Lambda} = \mathbf{S}^2$. Using this decomposition, the inverse of $\boldsymbol{\Sigma}$ in Equation (1) can be computed efficiently as $\boldsymbol{\Sigma}^{-1} = \mathbf{V}(\alpha\mathbf{I} + \beta\boldsymbol{\Lambda})^{-1}\mathbf{V}^{\top}$, which avoids explicitly inverting a large dense matrix.

# 3  Federated Data Preprocessing

In this section, we present FedPS, a framework for federated preprocessing, which aims to estimate global preprocessing parameters from distributed local datasets. Different preprocessors require different sufficient statistics, which in turn determine communication costs. These parameters can be computed either exactly or approximately, depending on the communication overhead. For some statistics (e.g., means and variances), exact aggregation requires communication independent of local dataset size, yielding parameters identical to centralized computation. However, for other statistics (e.g., quantiles and frequent items), exact computation requires communication that scales with local dataset size, making it impractical. In such cases, sketch-based procedures reduce communication costs while providing explicit approximation guarantees. We first formulate the problem of federated preprocessing, and then present a general framework for federated preprocessing. We also describe representative preprocessing methods and analyze their communication costs.

## 3.1  Problem Formulation

Based on the above discussion, we formulate the problem of Federated Data Preprocessing as follows. We have a collection of $N$ clients, who each hold some data $X_i$. These clients collaborate with a server $\mathcal{S}$ in order to compute a function of interest, $F$. Specifically, we seek to find $F(\mathbf{X}) = F(\cup_{i=1}^{N} X_i)$. For many functions $F$, exact computation of $F$ may not be practical, and instead we seek an approximation, $\hat{F}$. The nature of the

Table 1: Preprocessors and associated statistics.

| Categories | Preprocessors | Formulation | Associated Statistics |
|---|---|---|---|
| Scaling | *MaxAbsScaler* | $x/|x|_{\max}$ | Max |
| | *MinMaxScaler* | $(x - x_{\min})/(x_{\max} - x_{\min})$ | Min, Max |
| | *StandardScaler* | $(x - \mu)/\sigma$ | Mean, Variance |
| | *RobustScaler* | $(x - Q_2)/(Q_3 - Q_1)$ | *Quantile* |
| | *Normalizer* | $x/\|x\|$ | Sum, Max |
| Encoding | *LabelBinarizer* | one-hot$(y)$ | Set Union |
| | *MultiLabelBinarizer* | multi-hot$(y)$ | Set Union |
| | *LabelEncoder* | ordinal$(y)$ | Set Union |
| | *OneHotEncoder* | one-hot$(x)$ | Set Union, *Frequent items* |
| | *OrdinalEncoder* | ordinal$(x)$ | Set Union, *Frequent items* |
| | *TargetEncoder* | $\lambda(n_i)\frac{n_{iY}}{n_i} + (1 - \lambda(n_i))\frac{n_Y}{n}$ | Set Union, Mean, Variance |
| Transformation | *PowerTransformer* | $\psi(\lambda, x)$ | Sum, Mean, Variance |
| | *QuantileTransformer* | CDF$(x)$, $\Phi^{-1}(\text{CDF}(x))$ | Quantile |
| | *SplineTransformer* | B-spline$(x)$ | Min, Max, *Quantile* |
| Discretization | *Binarizer* | 1 if $x > T$ else 0 | $-$ |
| | *KBinsDiscretizer* | $j$ if $T_j \leq x < T_{j+1}$ | Min, Max, Quantile, Mean |
| Imputation | *SimpleImputer* | mean$(x)$, median$(x)$, freq$(x)$ | Mean, *Quantile*, *Freq-items* |
| | *KNNImputer* | mean$(k$-NN of $x)$ | HFL: Min, Mean; VFL: Sum |
| | *IterativeImputer* | RegressionModel$(x)$ | Sum |

approximation will depend on the nature of $F$ and the approach taken. However, a generic goal is to minimize some norm of the deviation between $F$ and $\hat{F}$ for vector or scalar valued functions, $\|F(\mathbf{X}) - \hat{F}(\mathbf{X})\|_2$, e.g., the Euclidean norm. This can be viewed as a restricted instance of the more general machine learning objective of empirical loss minimization Jung (2022). Specific examples are described in the subsequent sections.

Throughout, we will seek to bound the *communication cost* of the protocol, in terms of both the worst-case amount of data communicated per client, and the number of rounds of interaction between the clients and the server. Consistent with other models of federated computation, we focus on protocols that do not require peer-to-peer communication between clients, and instead only allow communications that go between clients and the server. Rather than provide a fixed communication budget or bound on the allowable number of rounds, we analyze the results achievable for different methods, and note what tradeoffs are achievable. Our goal of communication efficiency is protocols whose cost is independent of the size of the data held by each client. That is, if client $i$ holds $n_i$ examples, their communication cost should be independent of $n_i$ (or at most very weakly dependent, i.e., $O(\log n_i)$). This may be achieved by adopting compact data summaries that can be combined by the server, such as sketch-based approximations. This means that the total amount of communication to the server scales linearly with the number of clients.

Our *privacy model* seeks to minimize the information that is shared between clients and the server (and thus is aligned with the objective to limit the communication cost). Thus trivial solutions which centralize all the client data at the server are prohibited. Instead, we seek solutions that adopt the principle of *data minimization*, and send messages that constitute summaries and aggregated statistics rather than raw data. Note that this does not absolutely guarantee that there is no leakage of information about the original data, and we do not make any claims in this regard. We discuss approaches to formalizing privacy guarantees in Section 6.

## 3.2 The Framework

Federated preprocessing follows the workflow in Figure 2: clients compute local statistics (Step ①), which are aggregated at the server (Step ②). The server derives preprocessing parameters (Step ③), broadcasts them to clients (Step ④), who apply them locally (Step ⑤). We outline representative preprocessors.

**Why these preprocessors?** We select these methods to span a broad range of statistical complexity and practical relevance in federated learning. These are core techniques implemented in standard machine learning libraries, particularly *Scikit-learn*, which makes them both familiar to practitioners and representative of real-world preprocessing pipelines. *StandardScaler* represents perhaps the most commonly used normalization technique and relies only on first- and second-order moments. *KBinsDiscretizer* captures discretization methods based on ranges, quantiles, and clustering, allowing us to illustrate the use of quantile sketches and federated $k$-Means. *KNNImputer* and *IterativeImputer* represent substantially more complex imputers that operate in both horizontal and vertical settings and rely on nontrivial federated primitives such as $k$-NN regression and Bayesian linear regression. Together, these methods demonstrate how FedPS supports preprocessing from simple aggregations to iterative, model-based procedures, while maintaining compatibility with widely-used preprocessing libraries.

*StandardScaler* computes a global mean and variance. Each client reports three quantities: the sum of squared values $s = \sum_i x_i^2$, the sum of values $c = \sum_i x_i$, and the number of samples $n$ (Step ①). The server aggregates these statistics by summation to obtain $(S, C, N)$ (Step ②). After aggregation, the global mean is $\mu = C/N$ and the variance is $\sigma^2 = S/N - \mu^2$ (Step ③). This requires one data pass and one communication round. The server then sends $\mu$ and $\sigma$ to clients (Step ④), who apply the transformation $(x - \mu)/\sigma$ (Step ⑤).

*KBinsDiscretizer* partitions continuous features into discrete bins using one of three strategies: uniform, quantile-based, or clustering-based. For uniform binning, clients compute local min/max values, which are aggregated at the server to obtain global bounds. The server sends these values to clients, who derive equal-width bin edges and use them for discretization. For quantile-based binning, clients construct local quantile sketches (Step ①), which the server merges to form a global sketch (Step ②). The resulting quantile-based bin edges are computed centrally (Step ③) and sent back to clients for discretization (Steps ④ and ⑤). For clustering-based binning, we employ federated $k$-Means (Appendix A.1). Clients iteratively contribute sufficient statistics of cluster assignments, while the server updates centroids by computing the global mean of points in each cluster. Final bin boundaries derived from the centroids are shared with clients.

*KNNImputer* fills missing values using the mean of the $k$ nearest neighbors, based on a distance function that accounts for missing coordinates and normalizes by the number of valid comparisons (Dixon, 1979). We implement this via federated $k$-Nearest Neighbors Regression (Appendix A.2). In the horizontal setting, each client computes distances between its local samples and query samples and reports its smallest $k$ distances to the server. The server aggregates these candidates to identify the global nearest neighbors and requests the corresponding feature values for imputation. In the vertical setting, distance computation is distributed across clients according to their feature subsets. Each client contributes partial distances, which the server aggregates and normalizes to determine the nearest neighbors used for imputation.

*IterativeImputer* treats each feature with missing values as a regression problem, predicting missing entries from all other features in an iterative, round-robin manner (Buck, 1960). This procedure resembles a simplified form of MICE (Buuren & Groothuis-Oudshoorn, 2011). The regression model is implemented using Federated Bayesian Linear Regression (Section 4). In each iteration, clients contribute the necessary second-order statistics, such as $\mathbf{X}^\top \mathbf{X}$ in the horizontal setting or $\mathbf{X}\mathbf{X}^\top$ in the vertical setting. The server aggregates these statistics to compute global regression parameters, which are then used to update missing values before proceeding to the next iteration.

Other preprocessing methods, including *MinMaxScaler*, *OrdinalEncoder*, *PowerTransformer*, and others, follow the same principle: identify the sufficient statistics, aggregate them across clients, compute global parameters at the server, and broadcast them back. For clarity, we group methods into five categories: scaling, encoding, transformation, discretization, and imputation. Table 1 summarizes the sufficient statistics for each method, with further details in Appendix B. *Italics statistics* denote sketch-based approximations.

**Data heterogeneity** is a common challenge in federated learning. Fortunately, the aggregated statistics we need are insensitive to distribution shifts among clients. Simple statistics such as minimum, maximum, sums, means, and variances are exact after aggregation. Quantile and frequent-item sketches maintain theoretical guarantees regardless of local distribution differences, making them well suited to heterogeneous environments.

### 3.3  Communication Overhead Analysis

Communication overhead is a critical element of federated learning. To assess the communication efficiency of different preprocessing methods, we analyze the statistics required by each method together with the resulting number of communication rounds and per client communication cost. Some preprocessors depend on a single aggregated statistic, while others require multiple statistics or iterative protocols. The total overhead is therefore determined by the combination of required statistics and their aggregation frequency.

We use the following parameters:

- $n$, $m$: number of samples and number of features in the training set.

- $n'$: number of samples in the test set.

- $t$: number of iterations in $k$-Means, power transform, and iterative imputation.

- $k$: method-dependent parameter such as number of clusters in $k$-Means, number of neighbors in $k$-Nearest Neighbors, or maximum map size in frequent-item sketch.

- $d$: number of distinct categories (for encoding tasks).

Table 2 reorganizes preprocessing methods by the type of aggregated statistic they require, rather than by functionality. This view complements Table 1, which identifies sufficient statistics for each method. Here, our goal is to make explicit how different statistical primitives translate into communication rounds and asymptotic costs per client under horizontal and vertical data partitioning. When a preprocessor relies on multiple statistics, its communication cost is the sum of the corresponding components.

Table 2 summarizes the communication cost of aggregating each class of statistics. Simple statistics such as minimum, maximum, sum, mean, and variance require a constant amount of communication per feature, resulting in $O(m)$ cost per client. *Normalizer* needs sums or maxima per row, which gives $O(n)$ total cost. More complex methods incur higher overhead due to sketch-based summaries, iterative procedures, or pairwise sample interactions, as exemplified by quantile estimation, clustering-based discretization, and $k$-nearest neighbor imputation. Overall, the table highlights how FedPS supports a wide range of preprocessing methods while making their communication requirements explicit and comparable.

Encoding methods based on set union depend on the number of distinct categories $d$, while *TargetEncoder* requires mean and variance per category, giving $O(d)$ communication cost. Frequent-item sketches incur a cost of $O(k)$ per feature, where $k$ is the maximum map size. The true frequency of an item lies between the Upper Bound (UB) and Lower Bound (LB) computed for that item, with $(\text{UB} - \text{LB}) \leq n\epsilon$, where $n$ denotes the sum of all item counts, and $\epsilon = 3.5/k$. The implementation of quantile-based methods relies on KLL sketches (Karnin et al., 2016), whose communication cost is governed by sketch size under error parameters $\epsilon$ and $\delta$. By default, the configuration targets $\epsilon \approx 1.65\%$. REQ sketches (Cormode et al., 2023) can also be used when relative-error guarantees are needed. By default, this setting roughly corresponds to $\epsilon \approx 1\%$.

For *KBinsDiscretizer* with $k$-means-based binning, the cost is $O(tkm)$. Quantile-based binning inherits the KLL sketch cost, while uniform binning requires only $O(m)$ communication since only min and max statistics are needed. For *KNNImputer*, the cost depends on the size of test set $n'$. In the horizontal setting, each client sends $O(n'k)$ distances per feature, and once the server identifies the nearest neighbors, the relevant clients send the associated $O(n'k)$ values in the worst case. This results in a cost of $O(n'km)$ per client. In the vertical setting, each client computes partial distances from all training samples to all test samples, yielding a cost of $O(n'n)$.

Table 2: Aggregated statistics and associated preprocessors.

| Statistics | Associated Preprocessors | Partitioning | Comm. Round | Comm. Cost (Client) |
|---|---|---|---|---|
| Min / Max | *MaxAbsScaler* | Horizontal | 1 | $O(m)$ |
| | *MinMaxScaler* | Horizontal | 1 | $O(m)$ |
| | *Normalizer* (max norm) | Vertical | 1 | $O(n)$ |
| | *KBinsDiscretizer* (uniform) | Horizontal | 1 | $O(m)$ |
| | *SplineTransformer* (uniform) | Horizontal | 1 | $O(m)$ |
| | *KNNImputer* | Horizontal | 1 | $O(n'km)$ |
| Sum | *Normalizer* ($l_1$ or $l_2$ norm) | Vertical | 1 | $O(n)$ |
| | *PowerTransformer* | Horizontal | 1 | $O(m)$ |
| | *KNNImputer* | Vertical | 1 | $O(n'n)$ |
| | *IterativeImputer* | Horizontal | $t$ | $O(tm^2 \min(n,m))$ |
| | *IterativeImputer* | Vertical | $t$ | $O(tmn(\min(n,m)+t))$ |
| Mean | *StandardScaler* ($\mu=0$) | Horizontal | 1 | $O(m)$ |
| | *SimpleImputer* (mean) | Horizontal | 1 | $O(m)$ |
| | *TargetEncoder* | Horizontal | 1 | $O(dm)$ |
| | *PowerTransformer* ($\mu=0$) | Horizontal | 1 | $O(m)$ |
| | *KBinsDiscretizer* (kmeans) | Horizontal | $t$ | $O(tkm)$ |
| | *KNNImputer* | Horizontal | 1 | $O(n'km)$ |
| Variance | *StandardScaler* ($\sigma=1$) | Horizontal | 1 | $O(m)$ |
| | *TargetEncoder* | Horizontal | 1 | $O(dm)$ |
| | *PowerTransformer* | Horizontal | $t$ | $O(tm)$ |
| Quantiles | *RobustScaler* | Horizontal | 1 | |
| | *KBinsDiscretizer* (quantile) | Horizontal | 1 | |
| | *QuantileTransformer* | Horizontal | 1 | $O(\frac{1}{\epsilon}\log^2\log\frac{1}{\delta}\cdot m)$ |
| | *SplineTransformer* (quantile) | Horizontal | 1 | |
| | *SimpleImputer* (median) | Horizontal | 1 | |
| Set Union | *LabelBinarizer* | Horizontal | 1 | |
| | *MultiLabelBinarizer* | Horizontal | 1 | $O(d)$ |
| | *LabelEncoder* | Horizontal | 1 | |
| | *OneHotEncoder* | Horizontal | 1 | |
| | *OrdinalEncoder* | Horizontal | 1 | $O(dm)$ |
| | *TargetEncoder* | Horizontal | 1 | |
| Freq-items | *OneHotEncoder* (ignore infreq.) | Horizontal | 1 | |
| | *OrdinalEncoder* (ignore infreq.) | Horizontal | 1 | $O(k \cdot m)$ |
| | *SimpleImputer* (most-frequent) | Horizontal | 1 | |

For *IterativeImputer*, the algorithm iterates over all features with missing values (worst case $m$) for $t$ rounds. In each round, one feature is treated as the target and regressed on all other features using Federated Bayesian Linear Regression. The communication cost per iteration depends on the sufficient statistics aggregated by BLR. For the horizontal setting, BLR costs $O(m \min(n,m))$ (Theorem 4.1), and for the vertical setting, it costs $O(n \min(n,m) + nt)$ (Theorem 4.3). Since *IterativeImputer* performs $t$ iterations over $m$ features, the total communication cost is $O(tm^2 \min(n,m))$ for the horizontal setting and $O(tmn(\min(n,m)+t))$ for the vertical setting.

## 4 Federated Bayesian Linear Regression

In this section we present a focused study of a specific preprocessing technique as an illustration of applying the FedPS framework to a more complex task. Bayesian Linear Regression (BLR) (Section 2.4) is a foundational

technique used within *IterativeImputer* to model conditional feature distributions. Unlike simpler preprocessing models such as $k$-Means, which uses only cluster sum and count, or $k$-Nearest Neighbors, which uses distances, BLR maintains a full posterior distribution over model parameters and iteratively refines hyperparameters $\alpha$ and $\beta$. This iterative refinement is essential for accurate imputation but introduces complexity absent in one-shot aggregation methods.

In federated settings, the key challenge is computing sufficient statistics for parameter updates without exposing raw data, while accommodating the distinct computational patterns of horizontal versus vertical data partitioning. We develop federated BLR for both settings to illustrate how FedPS extends beyond simple aggregations to support sophisticated model-based preprocessing with multiple communication rounds. The horizontal variant follows standard sufficient-statistic aggregation and serves as a baseline. The vertical variant introduces an algebraic reformulation that enables exact Bayesian inference without cross-client feature interactions. Both variants iteratively update model parameters until convergence.

### 4.1 Horizontal Federated BLR

---
**Algorithm 1** Horizontal Federated BLR (Server)

---
1: **Input:** Client $c$ holds $\mathbf{X}^{(c)}$ and $\mathbf{Y}^{(c)}$
2: Initialize $\alpha$ and $\beta$
3: Aggregate $\mathbf{X}^\top\mathbf{Y} = \sum_c \mathbf{X}^{(c)^\top}\mathbf{Y}^{(c)}$
4: Aggregate $\mathbf{X}^\top\mathbf{X} = \sum_c \mathbf{X}^{(c)^\top}\mathbf{X}^{(c)}$
5: Compute eigenvalues $\mathbf{\Lambda}$ and eigenvectors $\mathbf{V}$ of $\mathbf{X}^\top\mathbf{X}$
6: **repeat**
7:     Compute $\mathbf{\Sigma}^{-1} = \mathbf{V}(\alpha\mathbf{I} + \beta\mathbf{\Lambda})^{-1}\mathbf{V}^\top$
8:     Compute $\hat{\boldsymbol{\omega}} = \beta\mathbf{\Sigma}^{-1}\mathbf{X}^\top\mathbf{Y}$
9:     Broadcast $\hat{\boldsymbol{\omega}}$ to all clients
10:     Aggregate global error $\varepsilon = \sum_c \|\mathbf{Y}^{(c)} - \mathbf{X}^{(c)}\hat{\boldsymbol{\omega}}\|_2^2$
11:     Update $\alpha$ and $\beta$ using Equation (2)
12: **until** Convergence or maximum number of iterations reached
13: **Output:** model parameter $\hat{\boldsymbol{\omega}}$

---

In the horizontal setting, samples are partitioned across clients, and each client holds all features for its local data. BLR depends on the sufficient statistics $\mathbf{X}^\top\mathbf{Y}$ and $\mathbf{X}^\top\mathbf{X}$, both of which decompose additively across clients. Each client computes its local contributions $\mathbf{X}^{(c)^\top}\mathbf{Y}^{(c)}$ and $\mathbf{X}^{(c)^\top}\mathbf{X}^{(c)}$, which are aggregated by summation at the server (Algorithm 1, steps 3-4). For example, $\mathbf{X}^\top\mathbf{X}$ is computed as ($\mathbf{X}$ are rows stacked):

$$\mathbf{X}^\top\mathbf{X} = \begin{bmatrix} \mathbf{X}^{(1)^\top} & \mathbf{X}^{(2)^\top} & \dots \end{bmatrix}\begin{bmatrix} \mathbf{X}^{(1)} \\ \mathbf{X}^{(2)} \\ \vdots \end{bmatrix} = \sum_c \mathbf{X}^{(c)^\top}\mathbf{X}^{(c)} \tag{3}$$

This procedure is equivalent to centralized BLR and avoids iterative FedAvg-style averaging (McMahan et al., 2017), since the sufficient statistics are exact after aggregation. We include this formulation for completeness and as a reference point for comparison with the vertical setting.

To avoid repeatedly inverting the current covariance matrix $\mathbf{\Sigma}$ in each iteration, the server performs an eigenvalue decomposition of $\mathbf{X}^\top\mathbf{X}$ once: $\mathbf{X}^\top\mathbf{X} = \mathbf{V}\mathbf{\Lambda}\mathbf{V}^\top$ (step 5). This enables fast updates of $\mathbf{\Sigma}^{-1}$ in each iteration (step 7). The server then computes $\hat{\boldsymbol{\omega}}$ and broadcasts it to clients (step 8-9). The computation cost of this algorithm for the server is dominated by the $O(m^3)$ decomposition cost (as a function of the number of features $m$). In practice, the computation time for this step is quite low. For example, when $m = 1000$, the decomposition only takes around 80 ms. Clients compute their local reconstruction error contributions, which are summed at the server to obtain the global error $\varepsilon$ (step 10), followed by updates to $\alpha$ and $\beta$ using Equation (2).

---

**Algorithm 2** Vertical Federated BLR (Server)

---

1: **Input:** Client $c$ holds feature block $\mathbf{X}^{(c)}$; one client holds $\mathbf{Y}$
2: Initialize $\alpha$ and $\beta$
3:   Receive $\mathbf{Y}$ from the client holding the target
4:   Aggregate $\mathbf{X}\mathbf{X}^\top = \sum_c \mathbf{X}^{(c)}\mathbf{X}^{(c)\top}$
5: Compute eigenvalues $\mathbf{\Lambda}$ and eigenvectors $\mathbf{U}$ of $\mathbf{X}\mathbf{X}^\top$
6: **repeat**
7:     Compute $\check{\mathbf{\Sigma}}^{-1} = \mathbf{U}(\alpha\mathbf{I} + \beta\mathbf{\Lambda})^{-1}\mathbf{U}^\top$
8:     Broadcast $\beta\check{\mathbf{\Sigma}}^{-1}\mathbf{Y}$ to all clients
9:     Each client computes $\hat{\boldsymbol{\omega}}^{(c)} = \mathbf{X}^{(c)\top}\beta\check{\mathbf{\Sigma}}^{-1}\mathbf{Y}$
10:     Aggregate $\hat{\mathbf{Y}} = \sum_c \mathbf{X}^{(c)}\hat{\boldsymbol{\omega}}^{(c)}$
11:     Compute error $\varepsilon = \|\mathbf{Y} - \hat{\mathbf{Y}}\|_2^2$
12:     Aggregate $\|\hat{\boldsymbol{\omega}}\|_2^2 = \sum_c \|\hat{\boldsymbol{\omega}}^{(c)}\|_2^2$
13:     Update $\alpha$ and $\beta$ using Equation (2)
14: **until** Convergence or maximum number of iterations reached
15: **Output:** model parameter $\hat{\boldsymbol{\omega}}$

---

The full procedure is outlined in Algorithm 1, with blue highlighted steps indicating communication from clients to server, and orange highlighted steps indicating communication from server to clients. In each iteration, clients communicate only scalar errors, making the iterative refinement phase communication-efficient (i.e., each client only has to send a single scalar). The dominant cost comes from the initial aggregation of sufficient statistics.

**Theorem 4.1.** *The communication cost per client of Horizontal Federated BLR is $O(m\min(n, m))$.*

*Proof.* Since $\mathbf{X}^{(c)}$ has size $n \times m$, in step 3 clients communicate $\mathbf{X}^{(c)\top}\mathbf{Y}^{(c)}$ of size $O(m)$. In step 4, the aggregate $\mathbf{X}^{(c)\top}\mathbf{X}^{(c)}$ is $O(m^2)$, but can be reduced to $O(m\min(n, m))$ via local eigenvalue decomposition. During iterative refinement, clients communicate only scalar errors at step 10, which is negligible. Thus, the dominant cost is the initial aggregation of $\mathbf{X}^{(c)\top}\mathbf{X}^{(c)}$, yielding $O(m\min(n, m))$ per-client communication. $\qquad\square$

### 4.2 Vertical Federated BLR

In the vertical setting, features are partitioned across clients while samples are aligned. Standard BLR requires $\mathbf{X}^\top\mathbf{X}$, whose off-diagonal blocks $\mathbf{X}^{(j)\top}\mathbf{X}^{(k)}$ $(j \neq k)$ involve cross-client feature interactions that cannot be computed without sharing raw data.

$$\mathbf{X}^\top\mathbf{X} = \begin{bmatrix} \mathbf{X}^{(1)\top} \\ \mathbf{X}^{(2)\top} \\ \vdots \end{bmatrix} \begin{bmatrix} \mathbf{X}^{(1)} & \mathbf{X}^{(2)} & \dots \end{bmatrix} = \begin{bmatrix} \mathbf{X}^{(1)\top}\mathbf{X}^{(1)} & \mathbf{X}^{(1)\top}\mathbf{X}^{(2)} & \dots \\ \mathbf{X}^{(2)\top}\mathbf{X}^{(1)} & \mathbf{X}^{(2)\top}\mathbf{X}^{(2)} & \dots \\ \vdots & & \end{bmatrix}. \tag{4}$$

To avoid this limitation, we reformulate the posterior mean computation using an equivalent expression based on $\mathbf{X}\mathbf{X}^\top$ instead of $\mathbf{X}^\top\mathbf{X}$, thereby avoiding cross-client terms:

$$\hat{\boldsymbol{\omega}} = \beta\mathbf{X}^\top\check{\mathbf{\Sigma}}^{-1}\mathbf{Y}, \quad \check{\mathbf{\Sigma}} = \alpha\mathbf{I} + \beta\mathbf{X}\mathbf{X}^\top. \tag{5}$$

Although $\check{\mathbf{\Sigma}}$ is not the posterior covariance, this reformulation yields the exact same posterior mean as the standard BLR solution. Crucially, $\mathbf{X}\mathbf{X}^\top$ decomposes additively across clients as $\sum_c \mathbf{X}^{(c)}\mathbf{X}^{(c)\top}$. Note that if we did want to output the posterior covariance, we would still require to compute $\mathbf{X}^\top\mathbf{X}$. However, since the preprocessing procedure of *IterativeImputer* uses only the posterior mean for imputation, this already satisfies our needs.

**Theorem 4.2.** *The posterior mean $\hat{\boldsymbol{\omega}}$ computed by the standard BLR formulation in Equation* (1) *is identical to that computed by the reformulated expression in Equation* (5).

*Proof.* To prove the equivalence of the two formulations, we show that $(\alpha\mathbf{I} + \beta\mathbf{X}^\top\mathbf{X})^{-1}\mathbf{X}^\top$ and $\mathbf{X}^\top(\alpha\mathbf{I} + \beta\mathbf{X}\mathbf{X}^\top)^{-1}$ are equal. Apply the Woodbury matrix identity (Golub & Van Loan, 2013) with $A = \alpha\mathbf{I}$, $U = \mathbf{X}^\top$, $V = \beta\mathbf{X}$. Then we have $(A + UV)^{-1}U = (\alpha\mathbf{I} + \beta\mathbf{X}^\top\mathbf{X})^{-1}\mathbf{X}^\top$. A corollary of the Woodbury identity gives $(A+UV)^{-1}U = A^{-1}U(I+VA^{-1}U)^{-1}$. Substituting and simplifying yields $A^{-1}U(I+VA^{-1}U)^{-1} = \alpha^{-1}\mathbf{X}^\top(\mathbf{I}+\beta\mathbf{X}\alpha^{-1}\mathbf{X}^\top)^{-1}$. Factoring $\alpha^{-1}$ inside the inverse cancels with the leading $\alpha^{-1}$, giving $\mathbf{X}^\top(\alpha\mathbf{I} + \beta\mathbf{X}\mathbf{X}^\top)^{-1}$, which completes the proof. □

We assume a server performs the aggregation and iterative refinement, while a single client holds the target vector, $\mathbf{Y}$, which is sent once to the server (Algorithm 2, step 3). The sufficient statistic required for BLR becomes $\mathbf{X}\mathbf{X}^\top = \sum_c \mathbf{X}^{(c)}\mathbf{X}^{(c)^\top}$. After aggregating this statistic (step 4), the server performs eigen-decomposition $\mathbf{X}\mathbf{X}^\top = \mathbf{U}\boldsymbol{\Lambda}\mathbf{U}^\top$ (step 5) to facilitate efficient computation of $\check{\boldsymbol{\Sigma}}^{-1} = \mathbf{U}(\alpha\mathbf{I} + \beta\boldsymbol{\Lambda})^{-1}\mathbf{U}^\top$ in each iteration (step 7). The updates of $\hat{\boldsymbol{\omega}}$ and $\check{\boldsymbol{\Sigma}}$ are tailored to the vertical setting as shown in Equation (5).

Since the features are partitioned across clients, each client only computes its local subset $\hat{\boldsymbol{\omega}}^{(c)}$. At each iteration, the server broadcasts $\beta\check{\boldsymbol{\Sigma}}^{-1}\mathbf{Y}$ (step 8), and each client computes its local coefficients $\hat{\boldsymbol{\omega}}^{(c)}$ (step 9). The server aggregates the global prediction $\hat{\mathbf{Y}}$ (step 10) and computes the error $\varepsilon$ (step 11), and aggregates the squared norm $\|\hat{\boldsymbol{\omega}}\|_2^2$ (step 12), which is required to update $\beta$. Algorithm 2 provides the full workflow.

**Theorem 4.3.** *The communication cost per client of Vertical Federated BLR is $O(n\min(n,m) + nt)$.*

*Proof.* In step 3, the client holding $\mathbf{Y}$ sends it to the server, which costs $O(n)$. In step 4, each client communicates $\mathbf{X}^{(c)}\mathbf{X}^{(c)^\top}$, which has size $O(n^2)$ but can be reduced to $O(n\min(n,m))$ via local eigen-decomposition. During the iterative refinement phase, at each iteration, clients communicate predictions in step 10 and parameter norms in step 12, each amounting to $O(n)$ per client. Across $t$ iterations, this contributes $O(nt)$ communication. Thus, the total per-client communication cost is dominated by the initial aggregation of $\mathbf{X}^{(c)}\mathbf{X}^{(c)^\top}$ plus iterative refinement, yielding $O(n\min(n,m) + nt)$. □

We note a potential privacy risk in step 3, where the client holding label $\mathbf{Y}$ sends it to the server in order to update the model parameter, which could potentially expose sensitive information. To mitigate this risk, we can employ cryptographic techniques such as secure multi-party computation (MPC) or homomorphic encryption to ensure that the server can compute the necessary statistics without directly accessing the raw labels. Additionally, if we assume that the server is the same entity as the client holding $\mathbf{Y}$, then step 3 could be omitted, as the server would already have access to the labels.

## 5 Empirical Evaluation

We study the impact of preprocessing in a federated learning environment and address some practical questions: (1) To what extent does preprocessing improve model performance? (2) How do different preprocessing choices behave under both IID and non-IID data partitions? (3) What are the actual communication costs per client of different preprocessing methods?

**Experiment Setup.** Referring back to the possible strategies outlined in Section 1, Centralized preprocessing (Option 1) is not feasible in realistic federated learning scenarios, and transfer preprocessing (Option 3) is not suitable when client distributions differ. Therefore, we evaluate three options: no preprocessing (Option 2), local preprocessing (Option 4), and federated preprocessing (Option 5). Experiments are conducted on three public tabular datasets: Adult (Becker & Kohavi, 1996), Bank (Moro et al., 2012), and Cover (Blackard, 1998). Twenty percent of each dataset is held out for testing, and the remaining eighty percent is used for training. Table 3 in the Appendix summarizes the dataset statistics.

We focus on two representative preprocessing techniques, *OrdinalEncoder* and *StandardScaler*, given their importance for tabular data where categorical features need encoding and feature magnitudes require

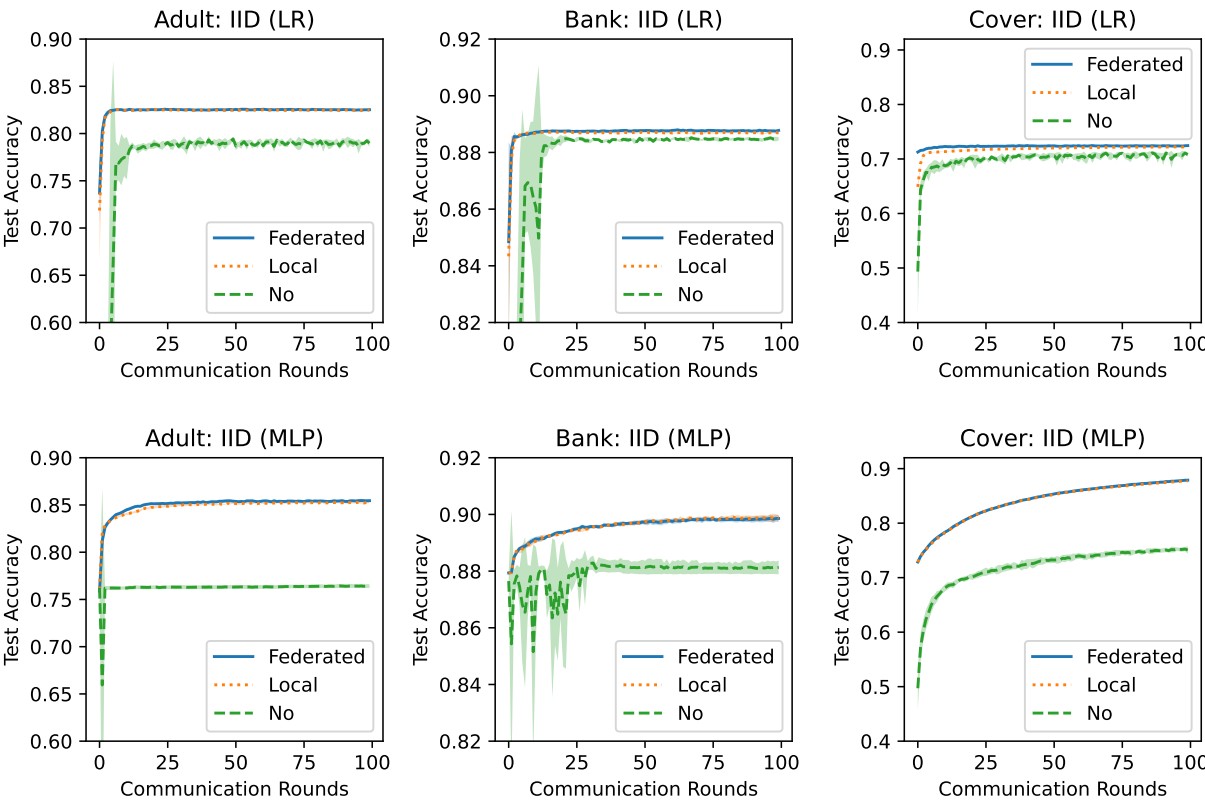

Figure 3: Test accuracy comparison in the IID setting (# Clients = 30).

normalization. For model training, we use FedAvg (McMahan et al., 2017) with the Adam optimizer (Kingma & Ba, 2015) to train Logistic Regression (LR) and a Multi Layer Perceptron (MLP) with two hidden layers of size 128 and 64. Each experiment runs for 100 communication rounds with one local epoch per round and a batch size of 32. The learning rate is tuned from $\{10^{-5}, 10^{-4}, 10^{-3}, 10^{-2}, 10^{-1}\}$, and results are averaged over five runs. We set the number of clients to 10 and 30, respectively, for the IID and non-IID settings. Code is available at `https://anonymous.4open.science/r/fl-tabular/README.md`.

**IID Data Partitioning.** In the IID setting, data is randomly shuffled and uniformly allocated across clients, so each client has a distribution close to the global distribution. Figure 3 and 6 (in the Appendix) show the test accuracy over communication rounds for the three preprocessing options, and Table 4 (in the Appendix) reports the final accuracies. Note that we include error bars in the plots (in the form of the shaded regions). However, since the variation is very small, so the error bars are often barely visible especially for the federated preprocessing option. Preprocessing consistently improves model performance across all datasets and models. On the Cover dataset with the MLP model, preprocessing increases accuracy by 17%. On the Adult dataset, the improvement is 5% for Logistic Regression and 12% for the MLP model. For the Bank dataset, the improvement is smaller, around 1 to 2%. As we would expect, under the IID setting, the local and federated preprocessing options behave similarly because local statistics align closely with global statistics. However, we note that, for both the IID and non-IID setting, the results obtained by the federated approach are *identical* to those that would be obtained if we were able to centralize all the data. The reason is that for these tasks, the federated statistics computed are mathematically equivalent to what we would compute centrally.

Note that on the Bank dataset, the improvement from preprocessing is smaller than on the other two datasets, roughly 1-2%. We think this is due to the characteristics of the dataset. First, most of the features in the Bank dataset are categorical, i.e., 9 out of 16 features are categorical, and the remaining 7 features are numerical. This reduces the impact of feature scaling relative to datasets with predominantly continuous

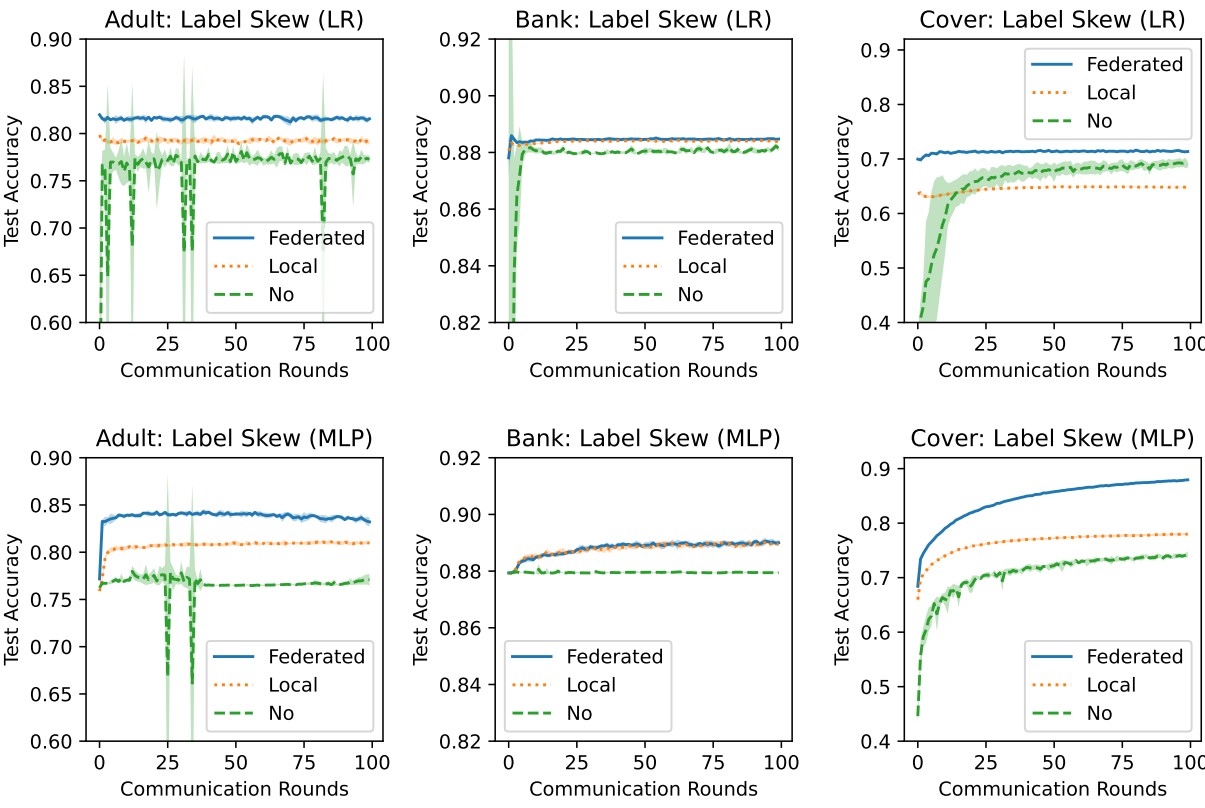

Figure 4: Test accuracy comparison in the label distribution skew setting (# Clients = 30).

features, e.g., the Cover dataset. Second, most categorical features have low cardinality (typically fewer than five categories), implying that local preprocessing and federated preprocessing produce highly similar encoding parameters even under heterogeneous data partitioning. Finally, although the improvement is smaller, we emphasize that preprocessing such as encoding is still necessary to convert categorical features into a numerical format suitable for model training.

**Non-IID Data Partitioning.** In the non-IID setting, data is partitioned using a label distribution skew and feature distribution skew strategies. For each client $j$ in the label skew setting, we sample $p_{k,j}$ from a Dirichlet distribution with parameter $\alpha = 0.5$ and assign a $p_{k,j}$ fraction of examples with label $k$ to that client. This creates heterogeneous label distributions, which is a common scenario in non-IID federated learning (Li et al., 2022). For the feature skew setting, we identify the continuous feature with the highest mutual information with the target label and generate feature skew by sorting the dataset on this feature prior to partitioning it. Results in Figures 4 and 5 (with additional Figures 7, 8 and Table 4 presented in the Appendix) show that preprocessing still provides substantial performance improvement over no preprocessing, especially on the Adult and Cover datasets. However, local preprocessing now performs appreciably worse than federated preprocessing. For instance, on the Cover dataset in label skew setting with the MLP model, local preprocessing yields an accuracy that is 11% lower than federated preprocessing. For Logistic Regression on the same dataset, the gap is 8%, which performs even worse than no preprocessing due to inconsistent local statistics. These results demonstrate that consistent preprocessing is essential for federated learning, particularly when client distributions differ.

**Communication cost.** To validate the communication cost analysis in Section 3.3, we measure the actual communication cost per client for different preprocessing methods on datasets with 1000 samples per client. Table 5 (in the Appendix) reports the total communication cost in kilobytes (KB). The results align with our theoretical analysis. *StandardScaler* incurs minimal overhead of approximately 0.6 KB per

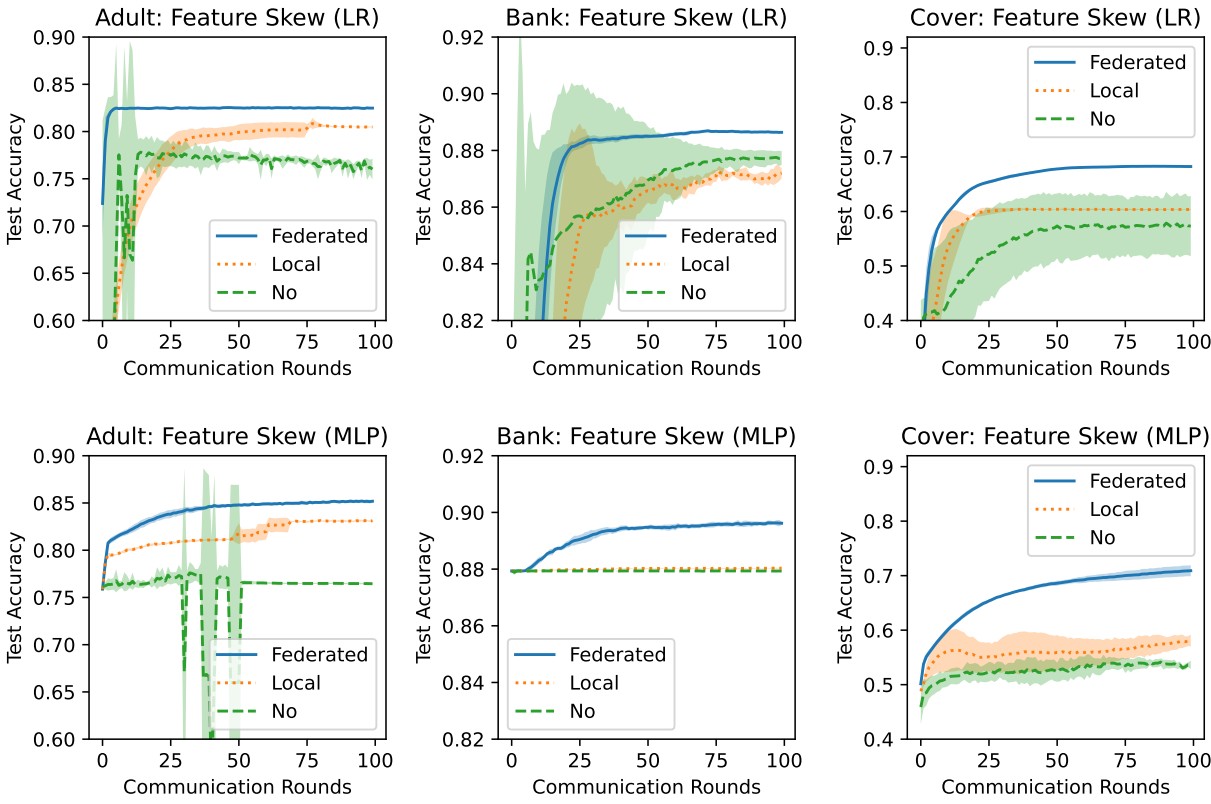

Figure 5: Test accuracy comparison in the feature distribution skew setting (# Clients = 30).

client on the Adult dataset, while *KBinsDiscretizer* with quantile-based and $k$-means-based binning incurs substantially higher costs of around 18 KB and 61 KB, respectively, due to quantile sketch aggregation and iterative clustering. Similarly, *SimpleImputer* with median and most-frequent strategies costs more than the mean strategy. For Federated Bayesian Regression in the horizontal setting, the Adult dataset incurs approximately 3 KB per client, reflecting the $O(m^2)$ cost of aggregating $\mathbf{X}^\top\mathbf{X}$, where $m$ is the number of features. For *PowerTransformer*, the communication cost is relatively high at around 73 KB, reflecting the need to iteratively optimize transformation parameters for each feature. Overall, these measurements confirm the communication cost estimates and demonstrate the trade-offs between preprocessing complexity and communication efficiency in federated learning.

**Consistency with Centralized Preprocessing.** Since the objective of FedPS is to faithfully reproduce centralized preprocessing in a federated setting, we evaluate each representative preprocessor by comparing the transformed dataset with those produced by the corresponding centralized implementation. Note that we exclude the features that are binary as they do not require any transformation. We compute the mean squared error (MSE) between the federated and centralized outputs, and report the results in Table 6 (in the Appendix). For most preprocessors, the MSE is negligible (less than 0.1), indicating that the federated implementation closely matches the centralized version. For those MSE equal to zero, the federated and centralized outputs are identical. For *SplineTransformer* using quantiles as the knots, the MSE is slightly higher, which is expected due to the polynomial transformation (degree of 3) applied to the quantile-based knots.

## 6 Discussion

**Existing work** on federated preprocessing is limited, as most frameworks focus on model training. In particular, some prominent toolkits, including *FATE* (Liu et al., 2021) and *SecretFlow* (The SecretFlow Authors, 2022), are primarily designed for secure multi-party model training and could potentially be extended to support a broader range of preprocessing methods. Baunsgaard et al. (2021; 2022) proposed federated data preparation methods but support only a limited set of preprocessors, such as *StandardScaler* and *KBinsDiscretizer* with uniform binning. Compared to existing frameworks, FedPS supports a wider range of preprocessing techniques with flexible, user-configurable parameters and explicit communication overhead analysis. For example, none of the existing implementations address dimensional explosion in *OneHotEncoder*, which we handle by restricting category counts and filtering low-cardinality items using frequent-item sketches. Table 7 in the Appendix summarizes the available options.

**Privacy** is another important aspect of federated preprocessing since sharing statistics can potentially leak information about local data. Prior work has explored secure preprocessing protocols, such as secure multi-party computation for the Yeo-Johnson transform (Marchand et al., 2022) and encoding schemes using fully homomorphic encryption (Hsu & Huang, 2022). Privacy guarantees depend on the deployment setting and adversarial assumptions. Different security models, such as honest-but-curious servers or colluding clients, can be supported using standard mechanisms including secure aggregation, secure multiparty computation, and trusted execution environments, since we have identified the sufficient statistics required for each preprocessing method. However, designing private protocols for complex statistics like quantiles requires careful consideration of trade-offs between latency, privacy, accuracy, and communication cost. Additionally, iterative preprocessing introduces practical challenges that complicate privacy protection, such as deciding whether intermediate results need to be kept private. Due to these complexities, we leave privacy-preserving preprocessing for future work.

## 7 Conclusion

This work highlights the essential yet often overlooked role of data preprocessing in federated learning. We introduce FedPS, a suite of tools that combines aggregated statistics, data sketches, and federated models. Experiments show that federated preprocessing substantially improves model accuracy, whereas inconsistent local preprocessing can reduce performance under non-IID data. Our framework targets the server-client architecture, the most widely adopted model for FL systems. Adapting FedPS to decentralized (peer-to-peer) settings would require additional mechanisms for direct client-to-client communication and consensus; see Jung (2026) for a discussion of network-structured FL beyond the server-client model.

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

# A Federated Machine Learning Models

## A.1 Federated k-Means

The $k$-Means clustering algorithm is an unsupervised method for identifying $k$ cluster centroids $\mu_1, \ldots, \mu_k$. Each iteration computes the distances between every data point and all centroids, assigns each point $x_i$ to the closest cluster $S_j$, and then updates each centroid as the mean of the points in that cluster: $\mu_j = \sum_{x_i \in S_j} x_i / n_j$, where $n_j$ is the number of points in cluster $S_j$.

In the horizontal federated setting, summarized in Algorithm 3, the server broadcasts the centroids to all clients. Each client locally assigns its data points to clusters and computes the sum and count of points in every cluster. The server aggregates these values to update the global centroids. The process repeats until convergence or until the maximum number of iterations is reached. Communication steps are color coded: blue for server receiving from clients, and orange for server sending to clients.

---
**Algorithm 3** Horizontal Federated $k$-Means (Server)

---
1: **Input:** Client $c$ holds $\{x_i^{(c)}\}$
2: Initialize centroids $\{\mu_1, \ldots, \mu_k\}$
3: **repeat**
4:     Broadcast centroids $\{\mu_1, \ldots, \mu_k\}$ to all clients
5:     Each client assigns its samples $x_i^{(c)}$ to the closest cluster $S_j$
6:     Aggregate local sums $\{s_1^{(c)}, s_2^{(c)}, \ldots, s_k^{(c)}\}$ where $s_j^{(c)} = \sum_{x_i^{(c)} \in S_j} x_i^{(c)}$ and and counts $\{n_1^{(c)}, \ldots, n_k^{(c)}\}$
7:     Update centroids $\mu_j = \sum_c s_j^{(c)} / \sum_c n_j^{(c)}$
8: **until** Convergence or reaching the maximum number of iterations
9: **Output:** Centroids $\{\mu_1, \ldots, \mu_k\}$

---

## A.2 Federated Nearest Neighbors Regression

The $k$-Nearest Neighbors ($k$-NN) regression algorithm predicts the value of a target variable $y$ by averaging the target values of the $k$ closest samples to a given point $x$, typically measured using the Euclidean distance. Weighted averages can also be used, where weights are inversely related to distances.

In the horizontal setting (Khedr, 2008), each client locally computes the $k$ smallest distances between the query point $x_p$ and its own data, then sends these distances to the server. The server merges the candidates to obtain the global top $k$ neighbors, and then requests the corresponding labels from clients. Algorithm 4 describes the full workflow.

---
**Algorithm 4** Horizontal Federated $k$-NN Regression (Server)

---
1: **Input:** Client $c$ holds data $\{x_i^{(c)}, y_i^{(c)}\}$; query point $x_p$
2: Broadcast $x_p$ to all clients
3: Collect local top-$k$ minimum distances $\{d_1^{(c)}, \ldots, d_k^{(c)}\}$ from each client
4: Compute global top $k$ distances and identify their indices
5: Send the selected indices to the corresponding clients
6: Collect the corresponding values of $y^{(c)}$ and compute a (weighted) mean $\mu$
7: **Output:** Predicted value $\mu$

---

In the vertical setting, different clients hold different features of the same samples. Each client computes a partial distance contribution, which the server aggregates to compute the full distance. The server then

identifies the global $k$ nearest neighbors and sends their indices to the client responsible for prediction. The procedure is shown in Algorithm 5.

---

**Algorithm 5** Vertical Federated $k$-NN Regression (Server)

---

1: **Input:** Client $c$ holds data $\{x_i^{(c)}\}$ and query point $x_p^{(c)}$; one client holds the target $\{y_i\}$
2: Each client computes partial distance between $x_p^{(c)}$ and all samples
3:  Aggregate all partial distances to obtain full distances
4: Select global top $k$ distances and their indices
5:  Send indices to the client that holds the target values
6: That client computes the (weighted) mean $\mu$ of the corresponding targets
7: **Output:** Predicted value $\mu$

---

## B  Federated Data Preprocessors

**Scaling** adjusts data to a specific range before model training. Common techniques include:

- *MaxAbsScaler* scales each feature so its maximum absolute value is one. It only requires the global maximum absolute value $|x|_{\max}$. The scaling rule is $x/|x|_{\max}$.

- *MinMaxScaler* maps data to a target interval, typically $[0, 1]$. This requires the global minimum $x_{\min}$ and maximum $x_{\max}$. The default rule is $(x - x_{\min})/(x_{\max} - x_{\min})$.

- *RobustScaler* uses quantiles, such as the lower quartile $Q_1$ (0.25), the median $Q_2$ (0.5), and the upper quantile $Q_3$ (0.75), to reduce sensitivity to outliers. Clients send quantile sketches, which are aggregated to obtain global quantiles. The scaling rule is $(x - Q_2)/(Q_3 - Q_1)$.

- *Normalizer* rescales each data sample to have unit norm ($l_1$, $l_2$, or max norm). In horizontal federation, each client normalizes its samples independently. In vertical federation, the global norm of each sample must be computed by summing partial norms (for $l_1$ and $l_2$) or taking the maximum (for max norm) across clients, then dividing each feature value by the global norm, i.e., $x/\|x\|$.

**Encoding** categorical values into numerical representations is crucial for machine learning models.

- All encoders require computing the global union of categories.

- *LabelBinarizer*, *MultiLabelBinarizer*, and *LabelEncoder* are typically used for label encoding, which usually involves a single column.

- *OneHotEncoder* and *OrdinalEncoder* are used for feature encoding, often across multiple columns. They can optionally ignore infrequent categories or limit the number of output categories using a frequent-item sketch.

- *TargetEncoder* (Micci-Barreca, 2001) assigns a value to each category based on the distribution of the target $Y$. For binary label $Y$, the encoded value for category $i$ is $\lambda(n_i)\frac{n_{iY}}{n_i} + (1 - \lambda(n_i))\frac{n_Y}{n}$, where $n_i$ is the number of samples in category $i$, $n_{iY}$ is the number of samples in category $i$ with $Y = 1$, $n_Y$ is the global number of positives. The shrinkage parameter $\lambda(n_i) = \frac{n_i}{m+n_i}$ depends on the smoothing factor $m = \sigma_i^2/\tau^2$, where $\sigma_i^2$ is the within-category variance and $\tau^2$ is the global variance of $Y$. Thus, this encoder requires computing global per-category means and variances of the target.

**Transformations** apply nonlinear operations to reshape feature distributions.

- *PowerTransformer* is a parametric method that aims to make data more Gaussian. The parameter $\lambda$ is estimated by maximizing the log-likelihood, which requires global sums and variances of the

transformed data. After applying the transformation, *StandardScaler* is used to obtain zero mean and unit variance. Xu & Cormode (2026) further discusses federated implementations with improved numerical stability.

- *QuantileTransformer* is a non-parametric method that maps data to a Uniform or Gaussian distribution. For the uniform case, the transformation outputs the empirical CDF (cumulative distribution function) value. For the Gaussian case, it applies the inverse Gaussian CDF $\Psi^{-1}$ to that value. Both transformations require global quantiles, computed via a quantile sketch.

- *SplineTransformer* constructs B-spline bases (de Boor, 1978). It follows a procedure similar to *KBinsDiscretizer*, where knot positions are chosen uniformly using global minimum and maximum values or along the global quantiles.

**Discretization** converts continuous variables into discrete categories.

- *Binarizer* applies a fixed threshold and does not require any federated computation.

**Imputation** addresses missing values in datasets.

- *SimpleImputer* is a univariate method that replaces missing values with the feature mean, median, or the most-frequent value. Means require global sums and counts. Medians use the quantile sketch, and the most-frequent value uses the frequent-item sketch.

- *IterativeImputer* is a multivariate, machine learning model based imputation method and BLR is used as the predictive model. The base case is the single-column scenario, where only one column has missing values. In this case, missing values in this column are treated as prediction targets, while the remaining columns are used as features. A regression model is fitted using samples with observed values and then used to predict missing entries. The more complex variant is the iterative case, where missing values may occur in multiple columns. In this setting, each column is treated as the target in a round-robin fashion, iteratively refining the imputed values. BLR is the default predictive model because it empirically performs well for imputation tasks.

## C  Additional Tables

Table 3: Dataset statistics.

| Datasets | # Examples | # Features | # Categorical | # Classes |
|----------|-----------|-----------|---------------|-----------|
| Adult | 33K | 14 | 8 | 2 |
| Bank | 45K | 16 | 9 | 2 |
| Cover | 581K | 54 | 0 | 7 |

Table 4: Test accuracy comparison of FedAvg using Logistic Regression (LR) and Multi-Layer Perceptron (MLP) in IID and non-IID (label and feature distribution skew) settings with different preprocessing options.

| # Clients | Partition | Model | Preprocessing | Adult | Bank | Cover |
|---|---|---|---|---|---|---|
| 10 | IID | LR | No | $0.79 \pm 0.0037$ | $0.88 \pm 0.0009$ | $0.71 \pm 0.0013$ |
| | | | Local | $0.83 \pm 0.0005$ | $0.89 \pm 0.0004$ | $0.72 \pm 0.0002$ |
| | | | Federated | $0.83 \pm 0.0003$ | $0.89 \pm 0.0002$ | $0.72 \pm 0.0011$ |
| | | MLP | No | $0.78 \pm 0.0094$ | $0.89 \pm 0.0009$ | $0.78 \pm 0.0049$ |
| | | | Local | $0.86 \pm 0.0007$ | $0.90 \pm 0.0009$ | $0.90 \pm 0.0013$ |
| | | | Federated | $0.86 \pm 0.0016$ | $0.90 \pm 0.0015$ | $\mathbf{0.91} \pm 0.0013$ |
| | Label skew | LR | No | $0.79 \pm 0.0048$ | $0.88 \pm 0.0005$ | $0.71 \pm 0.0030$ |
| | | | Local | $0.81 \pm 0.0014$ | $0.89 \pm 0.0002$ | $0.62 \pm 0.0007$ |
| | | | Federated | $\mathbf{0.82} \pm 0.0006$ | $0.89 \pm 0.0001$ | $\mathbf{0.72} \pm 0.0008$ |
| | | MLP | No | $0.77 \pm 0.0074$ | $0.88 \pm 0.0002$ | $0.79 \pm 0.0032$ |
| | | | Local | $0.83 \pm 0.0013$ | $0.89 \pm 0.0008$ | $0.79 \pm 0.0009$ |
| | | | Federated | $\mathbf{0.85} \pm 0.0026$ | $0.89 \pm 0.0011$ | $\mathbf{0.90} \pm 0.0004$ |
| | Feature skew | LR | No | $0.79 \pm 0.0136$ | $0.88 \pm 0.0001$ | $0.59 \pm 0.0586$ |
| | | | Local | $0.80 \pm 0.0003$ | $0.84 \pm 0.0002$ | $0.62 \pm 0.0001$ |
| | | | Federated | $\mathbf{0.83} \pm 0.0004$ | $\mathbf{0.89} \pm 0.0001$ | $\mathbf{0.68} \pm 0.0005$ |
| | | MLP | No | $0.79 \pm 0.0118$ | $0.88 \pm 0.0001$ | $0.56 \pm 0.0196$ |
| | | | Local | $0.82 \pm 0.0024$ | $0.88 \pm 0.0002$ | $0.62 \pm 0.0033$ |
| | | | Federated | $\mathbf{0.85} \pm 0.0012$ | $\mathbf{0.90} \pm 0.0008$ | $\mathbf{0.76} \pm 0.0032$ |
| 30 | IID | LR | No | $0.79 \pm 0.0040$ | $0.88 \pm 0.0006$ | $0.71 \pm 0.0039$ |
| | | | Local | $0.82 \pm 0.0004$ | $0.89 \pm 0.0002$ | $0.72 \pm 0.0001$ |
| | | | Federated | $\mathbf{0.83} \pm 0.0003$ | $0.89 \pm 0.0001$ | $0.72 \pm 0.0006$ |
| | | MLP | No | $0.76 \pm 0.0020$ | $0.88 \pm 0.0024$ | $0.75 \pm 0.0070$ |
| | | | Local | $0.85 \pm 0.0009$ | $0.90 \pm 0.0011$ | $0.88 \pm 0.0012$ |
| | | | Federated | $0.85 \pm 0.0008$ | $0.90 \pm 0.0012$ | $0.88 \pm 0.0009$ |
| | Label skew | LR | No | $0.77 \pm 0.0041$ | $0.88 \pm 0.0009$ | $0.69 \pm 0.0062$ |
| | | | Local | $0.79 \pm 0.0017$ | $0.88 \pm 0.0003$ | $0.65 \pm 0.0002$ |
| | | | Federated | $\mathbf{0.82} \pm 0.0022$ | $0.88 \pm 0.0001$ | $\mathbf{0.71} \pm 0.0014$ |
| | | MLP | No | $0.77 \pm 0.0062$ | $0.88 \pm 0.0002$ | $0.74 \pm 0.0060$ |
| | | | Local | $0.81 \pm 0.0006$ | $0.89 \pm 0.0003$ | $0.78 \pm 0.0017$ |
| | | | Federated | $\mathbf{0.83} \pm 0.0049$ | $0.89 \pm 0.0012$ | $\mathbf{0.88} \pm 0.0010$ |
| | Feature skew | LR | No | $0.76 \pm 0.0109$ | $0.88 \pm 0.0024$ | $0.57 \pm 0.0552$ |
| | | | Local | $0.80 \pm 0.0004$ | $0.87 \pm 0.0018$ | $0.60 \pm 0.0008$ |
| | | | Federated | $\mathbf{0.82} \pm 0.0001$ | $\mathbf{0.89} \pm 0.0002$ | $\mathbf{0.68} \pm 0.0010$ |
| | | MLP | No | $0.76 \pm 0.0006$ | $0.88 \pm 0.0001$ | $0.54 \pm 0.0074$ |
| | | | Local | $0.83 \pm 0.0010$ | $0.88 \pm 0.0002$ | $0.58 \pm 0.0109$ |
| | | | Federated | $\mathbf{0.85} \pm 0.0017$ | $\mathbf{0.90} \pm 0.0007$ | $\mathbf{0.71} \pm 0.0102$ |

Table 5: Communication cost per client for different federated preprocessors.

| Preprocessor | Adult | Bank | Cover |
|---|---|---|---|
| *StandardScaler* | 0.57 KB | 0.61 KB | 1.21 KB |
| *OrdinalEncoder* | 1.51 KB | 0.69 KB | 0 KB |
| *OrdinalEncoder* (ignore infreq.) | 1.91 KB | 0.84 KB | 0 KB |
| *TargetEncoder* | 24.89 KB | 16.39 KB | 0 KB |
| *PowerTransformer* | 73.46 KB | 75.18 KB | 360.18 KB |
| *KBinsDiscretizer* (uniform) | 0.48 KB | 0.51 KB | 1.11 KB |
| *KBinsDiscretizer* (quantile) | 18.88 KB | 21.53 KB | 55.05 KB |
| *KBinsDiscretizer* (kmeans) | 61.00 KB | 96.40 KB | 193.36 KB |
| *SimpleImputer* (mean) | 0.46 KB | 0.48 KB | 0.82 KB |
| *SimpleImputer* (median) | 18.40 KB | 21.02 KB | 70.83 KB |
| *SimpleImputer* (most-frequent) | 22.29 KB | 20.01 KB | 55.88 KB |
| Bayesian Regression (horizontal) | 3.19 KB | 3.69 KB | 25.21 KB |

Table 6: Mean Squared Error (MSE) of transformed data between federated and centralized preprocessing.

| Preprocessor | Adult | Bank | Cover |
|---|---|---|---|
| *MaxAbsScaler* | 0 | 0 | 0 |
| *MinMaxScaler* | 0 | 0 | 0 |
| *StandardScaler* | 0 | 0 | 0 |
| *RobustScaler* | $0.0095 \pm 0.0183$ | $0.0009 \pm 0.0008$ | $0.0014 \pm 0.0006$ |
| *Normalizer* | 0 | 0 | 0 |
| *OneHotEncoder* | 0 | 0 | 0 |
| *OrdinalEncoder* | 0 | 0 | 0 |
| *KBinsDiscretizer* (uniform) | 0 | 0 | 0 |
| *KBinsDiscretizer* (quantile) | $0.0238 \pm 0.0132$ | $0.0652 \pm 0.0413$ | $0.0743 \pm 0.0328$ |
| *KBinsDiscretizer* (kmeans) | 0 | 0 | 0 |
| *PowerTransformer* | 0 | 0 | 0 |
| *QuantileTransformer* (uniform) | $1.033\text{e-}5 \pm 1.769\text{e-}6$ | $1.582\text{e-}5 \pm 1.982\text{e-}6$ | $3.131\text{e-}5 \pm 4.906\text{e-}6$ |
| *QuantileTransformer* (normal) | $0.0299 \pm 0.0166$ | $0.0638 \pm 0.0164$ | $0.0742 \pm 0.0135$ |
| *SplineTransformer* (uniform) | 0 | 0 | 0 |
| *SplineTransformer* (quantile) | $0.0406 \pm 0.0173$ | $0.2936 \pm 0.0567$ | $0.1103 \pm 0.0084$ |

Table 7: Support of federated data preprocessors across existing frameworks and FedPS.

| Preprocessor | FATE (v2.2.0) | SecretFlow (v1.14.0b0) | Baunsgaard et al. | FedPS |
|---|---|---|---|---|
| MaxAbsScaler | ✗ | ✗ | ✗ | ✓ |
| MinMaxScaler | ✓ | ✓ | ✓ | ✓ |
| StandardScaler | ✓ | ✓ | ✓ | ✓ |
| RobustScaler | ✗ | ✗ | ✗ | ✓ |
| Normalizer | ✗ | ✗ | ✗ | ✓ |
| LabelBinarizer | ✗ | ✗ | ✗ | ✓ |
| MultiLabelBinarizer | ✗ | ✗ | ✗ | ✓ |
| LabelEncoder | ✓ | ✓ | ✗ | ✓ |
| OneHotEncoder | ✓ | ✓ | ✓ | ✓ |
| OrdinalEncoder | ✗ | ✓ | ✓ | ✓ |
| TargetEncoder | ✗ | ✗ | ✗ | ✓ |
| PowerTransformer | ✗ | ✗ | ✗ | ✓ |
| QuantileTransformer | ✗ | ✗ | ✗ | ✓ |
| SplineTransformer | ✗ | ✗ | ✗ | ✓ |
| KBinsDiscretizer | ✓ | ✓ | ✓ | ✓ |
| SimpleImputer | ✓ | ✗ | ✓ | ✓ |
| KNNImputer | ✗ | ✗ | ✗ | ✓ |
| IterativeImputer | ✗ | ✗ | ✗ | ✓ |
| **Coverage** | 6/18 | 6/18 | 6/18 | 18/18 |

Some preprocessors are renamed for consistency.

# D   Additional Figures

The following figures show the test accuracy comparison of FedAvg using Logistic Regression (LR) and Multi-Layer Perceptron (MLP) in IID and non-IID (label and feature distribution skew) settings with different preprocessing options. The number of clients is set to 10. The results show that federated preprocessing consistently improves model accuracy, while inconsistent local preprocessing can lead to performance degradation under non-IID data distributions. These findings highlight the importance of proper data preprocessing in federated learning scenarios.

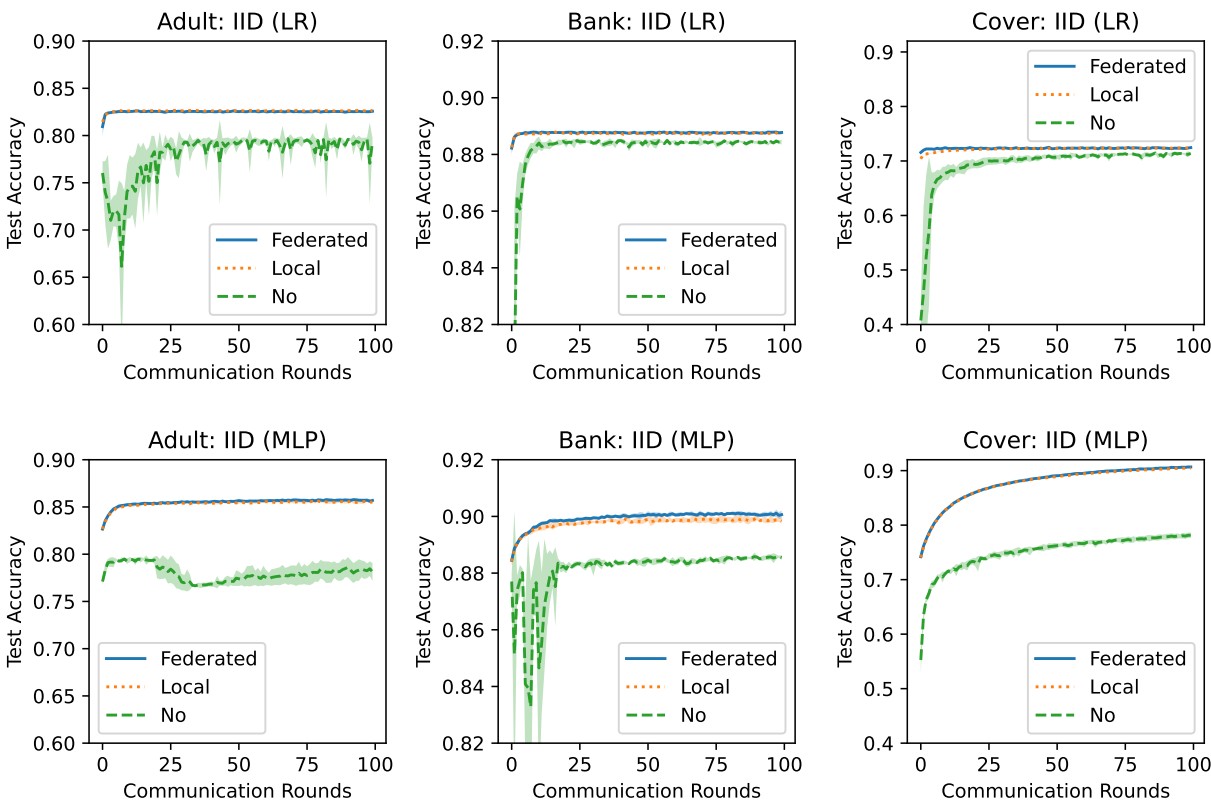

Figure 6: Test accuracy comparison in the IID setting (# Clients = 10).

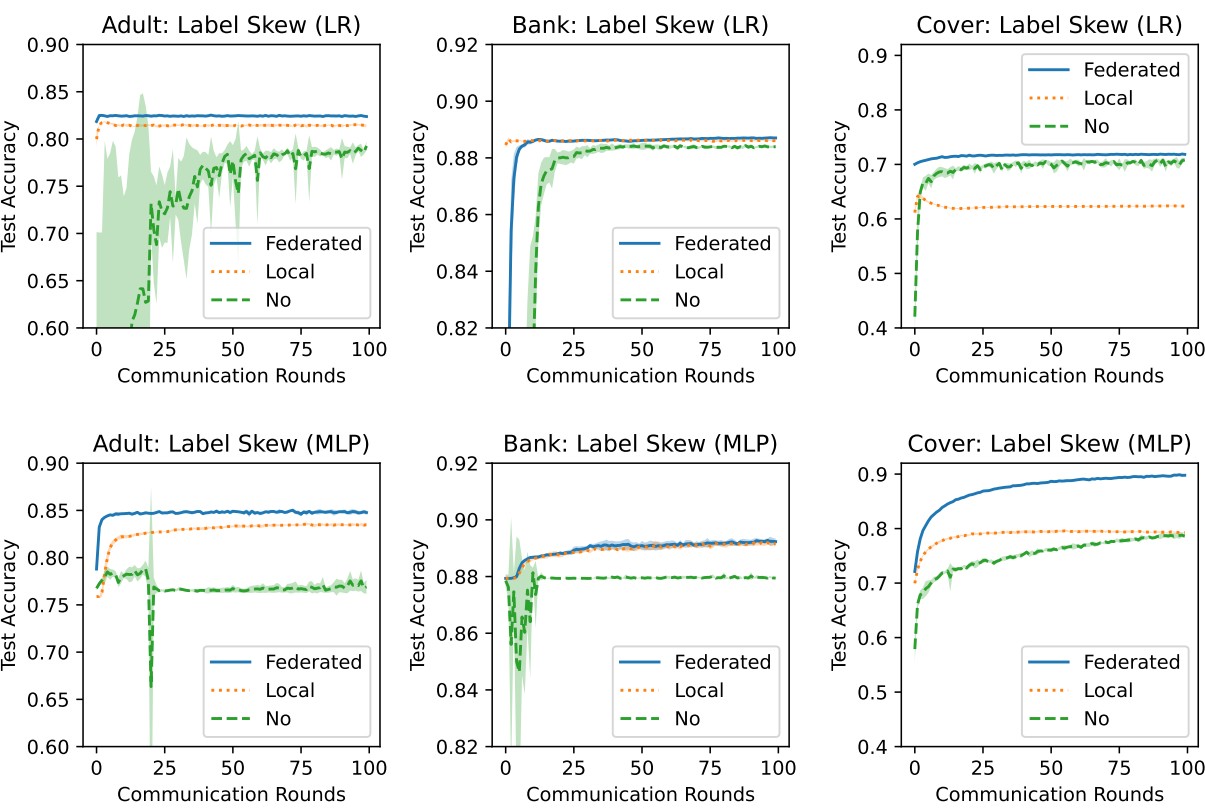

Figure 7: Test accuracy comparison in the label distribution skew setting (# Clients = 10).

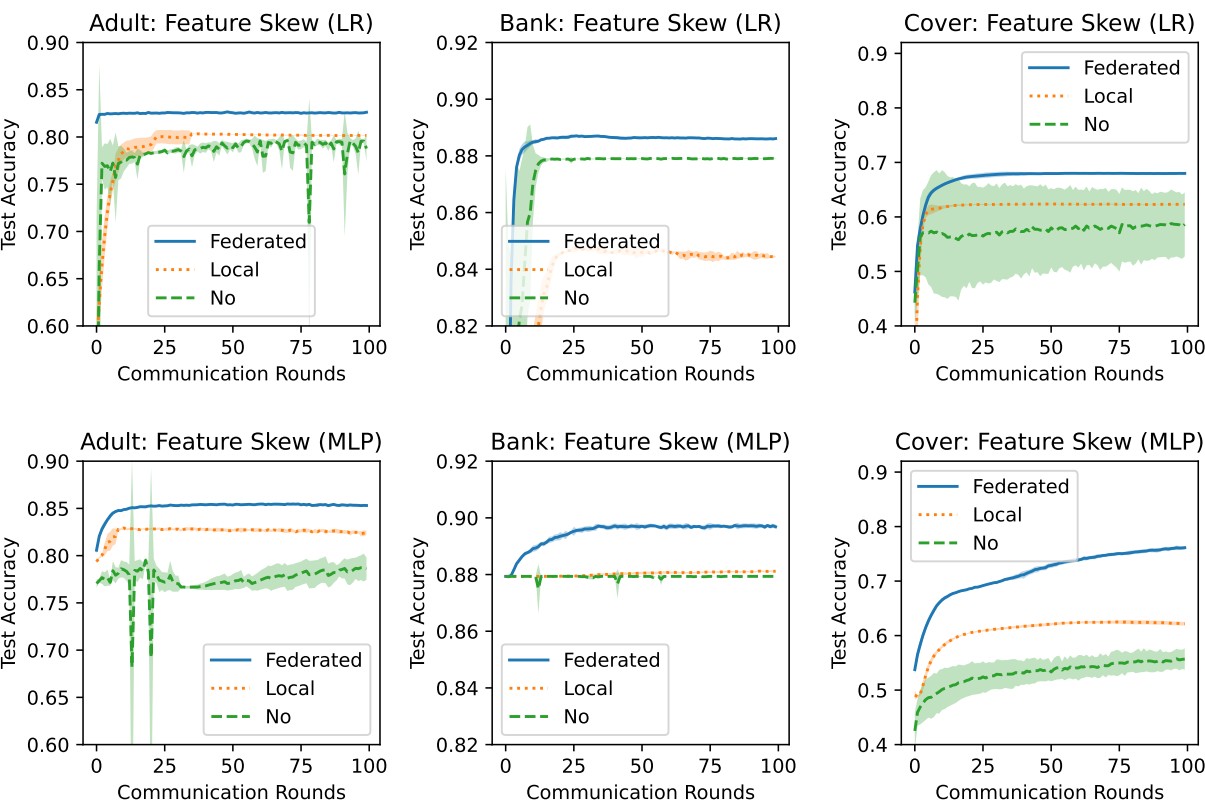

Figure 8: Test accuracy comparison in the feature distribution skew setting (# Clients = 10).

