# OpenReview forum: "FedPS: Federated Preprocessing for structured data via aggregated Statistics"
_TMLR — Under review for TMLR_

### Review · Reviewer_dUbV · 2026-04-01

**Summary Of Contributions:**

The paper makes the following contributions
- A conceptual framework and accompanying python library to implement federated data pre-processing.
- Experiments to study the impact of federated preprocessing vs no/local preprocessing.
- The development of a federated Bayesian Linear Regression algorithm in both horizontal and vertical scenario.

The authors adapt existing centralized data pre-processing techniques by leveraging established data sketching techniques to minimize communication complexity between the server and the client.

Strengths:
- Overall the paper is clear and well written. The motivations are clear and motivated by examples.
- The library is provider, and adopts the scikitlearn preprocessing API which is very convenient

Weaknesses:
- The algorithmic developments on Federated Bayesian Linear Regression feel disconnected from the rest of the paper.
- The experiments do not report results for all the implemented pre-processor which make it hard to evaluate the library just with this paper.

**Additional Comments:**

**[A1]** In table 2, you describe communication cost. In the paper you also mention that these methods have controllable approximation error compared to centralized pre-processing. Could that approximation error be added to table 2?

**[A2]**  One motivation of FL (and decentralized data processing) mentioned in the paper is privacy. Are there works on the privacy leakage on the federated pre-processors you mention?

**[A3]** In Table 4, data pre-processing does not seem to have an impact on the Bank results, do you have an idea why?

**Audience:**

Yes

**Audience Explanation:**

Data preprocessing is often overlooked even in centralized ML papers. The paper shines light on the data preprocessing step in a federated learning context an proposes an extensive library with baselines for federated preprocessing, it definitely has an audience.

**Claims And Evidence:**

Yes

**Claims Explanation:**

**[C1]** Local pre-processing is not suitable to federated learning, especially when client data is heterogeneous
- The thought-experiment of figure 1 helps illustrate that claim.
- Figure 3 and 4 on accuracy vs #communication rounds without (fig 3) and with (fig 4) biased label distribution clearly show that no/local pre-processing hurt performance.


**[C2]** FedPS easy to use and more complete than existing federated pre-processing libraries
- The scikit-learn compatible interface for the pre-processors make it easy to reuse the code.
- Table 6 exposes how FedPS supports more preprocessors than existing libraries


**[C3]** FedPS is expressive enough to adapt advanced data preprocessing techniques
- Section 4 shows an example on how to implement Bayesian Linear Regression in a federated setting.
- The code contains a lot of preprocessing, including learning based data imputation methods

**Requested Changes:**

**[R1]**  I believe that for an audience not familiar with Bayesian Linear Regression for data imputation, it is a bit unclear what BLR has to do with data imputation and preprocessing in general. Could you detail how it is used for preprocessing?

**[R2]**  To support the claim that the type of pre-processing is important for the performance of FL, could you add a comparison with the (theoretical) situation when preprocessing is centralized in fig 3 and 4?

---

> ### Author Response · Authors · 2026-07-09
>
> We thank the reviewer for this feedback.
>
> > The experiments do not report results for all the implemented pre-processor which make it hard to evaluate the library just with this paper.
>
> We note that several of the pre-processors are very basic, with exact implementations, (e.g., Normalizer, MinMaxScaler), and so it is not very informative to present extensive experiments on these.  In order to ensure that all the implemented methods have some empirical study, we have added Table 6 that reports the MSE of transformed data between federated and centralized preprocessing. The results demonstrate that the approximation methods are very close to the (theoretical) situation when preprocessing is centralized. Meanwhile, in Table 5, we report the communication cost on the client side.
>
> > Details of using Bayesian Linear Regression for data imputation
>
> We have expanded the explanation of the role of Bayesian Linear Regression (BLR) in preprocessing in Appendix B. BLR is used as the predictive model within imputation methods:
>
> Base case: Missing values in one column are treated as prediction targets, while the remaining columns are used as features. A regression model is fitted using samples with observed values and then used to predict missing entries.
>
> Iterative case: Missing values may occur in multiple columns. In this setting, each column is treated as the target in a round-robin fashion, iteratively refining the imputed values. BLR is the default predictive model because it empirically performs well for imputation tasks.
>
> > add a comparison with the (theoretical) situation when preprocessing is centralized in fig 3 and 4
>
> We have added more commentary on this point to the paper.  We observe that for the preprocessing methods evaluated in Figures 3–4 (OrdinalEncoder and StandardScaler), federated preprocessing produces the same result as centralized preprocessing because the required sufficient statistics are aggregated exactly. Specifically, OrdinalEncoder computes a global set union, while StandardScaler computes global means and variances. Therefore, there is no approximation error relative to centralized preprocessing in these cases.
>
> > add approximation error to table 2
>
> We have added the approximation error in section 3.3, also to differentiate between from exact computation, we have used italics to indicate that the estimates are approximate, as follows.
>
> KLL sketch (quantiles): a "single-sided" epsilon of about 1.33% and a "double-sided" (PMF) epsilon of about 1.65%.
>
> REQ sketch (quantiles): roughly corresponds to 1% relative error guarantee at 95% confidence.
>
> Frequent-item sketch: the true frequency will be between the Upper Bound (UB) and the Lower Bound (LB) computed for that item. Specifically, (UB- LB) ≤ n * epsilon, where n denotes the sum of all item counts, and epsilon=3.5/k, where k is the max Map Size.
>
> > Are there works on the privacy leakage on the federated pre-processors you mention
>
> Yes. We highlight related work in Section 6 that studies privacy risks in preprocessing procedures such as Yeo–Johnson transformations and encoding schemes. These works mention that sharing intermediate statistics can potentially reveal information about local client data.
>
> > In Table 4, data pre-processing does not seem to have an impact on the Bank results, do you have an idea why?
>
> We have added this discussion to section 5 (page 12). We think this behavior is due to the characteristics of the Bank dataset. Since most features are categorical, the impact of feature scaling is not prominent.

---

### Review · Reviewer_zChu · 2026-05-08

**Summary Of Contributions:**

The paper proposes a framework for federated data preprocessing.

Strengths
- Targets an under-served corner of FL (preprocessing, not yet another optimizer).
- Tables 1-2 (preprocessor -> sufficient statistic -> communication cost) are a useful reference.

Weaknesses

Main critique: no problem formulation.
The paper has a motivation (preprocessing matters in FL) and a
protocol (the five-step Compute -> Share -> Derive -> Broadcast ->
Apply workflow of Sec. 3.1), but never states the problem the
protocol is the solution to. Specifically, the paper does not
write down:

* a formal objective. given clients {(X_i, y_i)}, an admissible class of preprocessing maps T, what quantity
        is being optimized?
* a privacy / threat model. "privacy constraints prohibit centralizing raw data" is invoked as motivation,
        but i cannot find discussion of adversary, leakage bound, or formal guarantees.
* a communication model (with a budget B, a bound on rounds)
* a definition of "consistency
*  a definition of the estimand: each preprocessor (scaler,
        encoder, quantile, target encoder) approximates some global
        quantity from local samples; the population target and the
        approximation error are never written down.

The five-step workflow is a *protocol*, not a *problem*. Overall,
the paper lacks rigor:

- "Federated preprocessing significantly surpasses local and raw
  baselines" surpasses on what metric, defined how?
- "Communication-efficient" -> against what budget?
- "Consistent" -> in what norm, with what bound?
- "Privacy-preserving" -> against what adversary, with what
  guarantee?

Concrete request: open Sec. 3 with a problem formulation/setting.

The three-component framing (data, hypothesis
space, loss) in Jung, "Machine Learning: The Basics" (Springer, 2022)
might offer a template for specifying what each preprocessor
approximates and what the protocol may reveal.

Other weaknesses:
- Limited novelty: "compute sufficient statistics locally,
  sum at server" looks very much like folklore since FedAvg.
- Restricted to star-topology (server-client) FL. The five-step
  workflow assumes a central server that aggregates statistics and
  broadcasts parameters; this excludes decentralized / peer-to-peer
  FL where clients communicate over a graph without a central
  coordinator (gossip protocols, GTV-minimization over client
  networks, fully decentralized SGD). See, e.g., Jung,
  "Federated Learning: From Theory to Practice", Springer, 2026.
for a treatment
  of network-structured FL beyond the star topology. The paper
  should either (a) state explicitly that FedPS targets only the
  server-client setting, or (b) discuss which preprocessors
  generalize to decentralized aggregation.

- Privacy is the stated motivation but never analyzed (no threat
  model, no leakage discussion for X^T X).

- more empirical comparison against FATE/SecretFlow

**Audience:**

Yes

**Audience Explanation:**

FL preprocessing seems under-served, and a sklearn-style library is a useful artifact for
practitioners and benchmark builders.

**Claims And Evidence:**

No

**Claims Explanation:**

Federated preprocessing significantly surpasses local/raw baselines":
   shown only for two preprocessors (OrdinalEncoder + StandardScaler) on
   three small tabular datasets, with Dirichlet(0.5) label skew as the sole
   non-IID regime. Feature skew is not tested.

* No error bars; Table 4 reports point estimates over 5 runs without variance.

* "Communication-efficient" is asserted but the analysis is per-client: it hides
server-side compute (eigendecomposition of X^T X is O(m^3)), ignores scaling in
the number of clients C, and reports no wall-clock numbers.


* "Privacy prohibits centralizing raw data" motivates the work, yet the
  framework shares X^T X, quantile sketches, target means, and frequent items
  with no threat model. X^T X alone enables substantial reconstruction.

* Thm 4.2 establishes equivalence of the posterior mean only; calling the vertical
  procedure "Bayesian" seems misleading.

* Comparison to prior frameworks is a feature-coverage table (6) only.
 no head-to-head accuracy or cost numbers.

**Requested Changes:**

* Formal problem formulation. Define federated preprocessing as
  an estimation problem: given local data, what global parameter
  theta is being approximated, with what error, and with what
  guarantees ?  Ch 2 of Jung, "Machine Learning: The Basics" (Springer, 2022)
  offers a clean template for the formulation of ML applications.

* Threat model + leakage discussion. State the adversary
   (honest-but-curious server? colluding clients?)

Strengthening:

* One head-to-head comparison against FATE/SecretFlow (accuracy + communication).

* Test feature-distribution skew, not only label skew.

* Add error bars in Figs. 3-4 and stds in Table 4.

* Polish the language and avoid vague jargon:
   "unified framework"
    "straightforward sufficient statistics"
    "consistent preprocessing pipelines"
    "communication-efficient"
    "more versatile and comprehensive"
    "significantly broader"
    "more robust and efficient federated systems"
    with precise statements (which statistic, what error,
    what cost, on which baseline).

---

> ### Author Response · Authors · 2026-07-09
> **Comment Part 1**
>
> We thank the reviewer for this feedback.
>
> > a communication model / a formal objective / a privacy model.
>
> We have added Section 3.1 to more formally state the requirements of our federated setting and to better define the problem and the objectives.  We consider a number of different tasks, and there is not a “one size fits all” approach to approximation quality and communication cost.
>
> The objective of federated preprocessing is to estimate global preprocessing parameters from distributed local datasets. Depending on the required sufficient statistics, these parameters may be computed either exactly or approximately.
>
> For exact aggregation, the estimated parameters coincide with centralized computation (i.e., we observe zero change in utility for federated vs. centralized computation). For approximate methods, such as sketch-based procedures, the approximation error has explicit guarantees (Section 3.3). Our notion of communication efficiency refers to reducing this cost through sketch-based approximations.
>
> The privacy model is data minimization, since our primary focus is on the algorithmic perspective and limit the communication cost. We have added a more detailed discussion in Section 6.
>
> > "Communication-efficient" is asserted but the analysis is per-client: it hides server-side compute (eigendecomposition of X^T X is O(m^3)), ignores scaling in the number of clients C, and reports no wall-clock numbers.
>
> We note that in all our protocols, the total communication scales linearly with the number of clients. The eigen decomposition is performed once and avoids repeated matrix inversions. This cost is also present in centralized Bayesian regression and is not introduced uniquely by our framework. In practice the central cost can be negligible: when m=1000, the decomposition only takes around 80 ms.
>
> > more empirical comparison against FATE/SecretFlow. Comparison to prior frameworks is a feature-coverage table (6) only. no head-to-head accuracy or cost numbers.
>
> We have revised the description in Section 6 to clarify that FATE and SecretFlow are broader secure-computation frameworks rather than preprocessing methods themselves. While a limited number of pre-processing methods have been so far exhibited within FATE and SecretFlow, the whole of FedPS could in principle be implemented within them.  As summarized in Table 6, most preprocessors that they currently support relied on exact aggregation of statistics. As such, there is nothing to be gained in measuring accuracy since there is no loss in preprocessing accuracy relative to centralized computation.  Likewise, the computational and communication overhead primarily depends on the specific secure-computation protocol employed and so are not informative.
>
> > a definition of "consistency”, in what norm, with what bound?
>
> We have defined consistency more precisely in Section 1. In our context, consistency refers simply to all clients applying identical preprocessing parameters. Local preprocessing does not guarantee consistency because preprocessing parameters are estimated independently from local datasets, which may be heterogeneous and therefore produce different transformations.
>
> > a definition of the estimand: each preprocessor (scaler, encoder, quantile, target encoder) approximates some global quantity from local samples; the population target and the approximation error are never written down.
>
> We have provided the sufficient statistics required by each preprocessor concisely in Table 1. For exact aggregation, the resulting estimates coincide with centralized computation and incur no approximation error. For approximate methods, such as quantile estimation and frequent-item estimation, the approximation error is controlled by user-specified parameters with explicit guarantees.
>
> > Restricted to star-topology (server-client) FL.
>
> We have revised the paper to state this more explicitly in Section 3.1 and Section 7. Our framework focuses on the server–client architecture, which remains the dominant deployment setting in practical FL systems. While methods that operate within the star topology could be adapted to other architectures, extension to decentralized settings would require replacing centralized aggregation with distributed aggregation protocols.  Since this is not our main focus, we leave it as an interesting direction for future work.
>
> > "Federated preprocessing significantly surpasses local and raw baselines" surpasses on what metric, defined how?
>
> We clarify that this comment refers to the comparison based on test accuracy.  The evidence to support this assertion is reported in Figures 3-8 and Table 4.

---

> ### Author Response · Authors · 2026-07-09
> **Comment Part 2**
>
> > Add error bars in Figs. 3-4 and stds in Table 4.
>
> In fact, all lines on Figures 3–4 already include error bars through the shaded regions corresponding to mean ± standard deviation. The variance for federated and local preprocessing settings is substantially smaller than for the no-preprocessing baseline, making the intervals visually less noticeable. We have also added standard deviations to Table 4 for consistency.
>
> > Thm 4.2 establishes equivalence of the posterior mean only; calling the vertical procedure "Bayesian" seems misleading.
>
> We have clarified that Theorem 4.2 establishes an equivalent formulation only for the posterior mean in the vertical setting (by replacing computation of X^T X with X X^T, thereby avoiding cross-client terms). Indeed, it is the case that no equivalent formulation exists for the posterior covariance, which still requires X^T X. However, since preprocessing only uses the posterior mean, our procedure remains valid for this purpose.
>
> > Test feature-distribution skew, not only label skew.
>
> We have added experiments under feature-distribution skew (Figure 5, 8 and Table 4) as suggested. Specifically, we identify the continuous feature with the highest mutual information with the target label and generate feature skew by sorting on this feature prior to partitioning the dataset. As we might anticipate, the results are consistent with the label distribution skew setting, where federated preprocessing shows improved results compared to no preprocessing an local preprocessing.
>
> > Polish the language and avoid vague jargon.
>
> We appreciate this suggestion and have revised the draft.

---

### Review · Reviewer_bv5B · 2026-06-25

**Summary Of Contributions:**

The paper proposes FedPS, a framework for federated preprocessing of tabular data using aggregated statistics and sketches. Clients compute local statistics, the server aggregates them into global preprocessing parameters, and clients apply the resulting transformations locally. The paper covers a broad set of preprocessing operations and provides communication-cost analyses for the corresponding statistics. It also presents federated versions of k-Means, k-NN regression, and Bayesian linear regression, mainly for preprocessing and missing-value imputation.

Preprocessing is an important part of FL, especially in non-IID settings. A tool that makes standard preprocessing methods available could be useful for practitioners.

However, the paper's broad claims are not fully supported by the empirical evidence. The novelty over existing federated preprocessing systems should be clarified, and the privacy assumptions need to be stated much more carefully and sooner.

**Audience:**

No

**Audience Explanation:**

A tool for preprocessing in FL would be of interest to part of the TMLR audience.

The comparison between local and federated preprocessing is also interesting, because it highlights a practical issue that is often overlooked in FL experiments. However, in the current version, the empirical evidence is too limited to fully support the paper's broad claims.

**Claims And Evidence:**

No

**Claims Explanation:**

## General comment on the contribution

The paper seems to sit between two types of contribution: a practical software tool that maps standard Sklearn preprocessing methods to FL, and a methodological paper deriving federated versions of preprocessing routines. I think the tool aspect could be valuable, but the paper does not emphasize it strongly enough. Instead, much of the technical content consists of deriving well-known aggregations over clients using linear-algebra.

I would like the authors to be more explicit about which federated formulations are original contributions of this paper and which ones were already known in SOTA. As currently written, the paper sometimes gives the impression that the overall idea of federated preprocessing via local metadata/statistics and global aggregation is new, whereas related systems appear to already implement parts of this idea.

## About the evaluation

The claims in the submission are broader than the evidence provided. The paper presents FedPS as a comprehensive federated preprocessing suite covering scaling, encoding, transformation, discretization, and imputation. However, the main downstream experiments evaluate only OrdinalEncoder and StandardScaler. Many of the more distinctive advertised methods, such as RobustScaler, QuantileTransformer, PowerTransformer, TargetEncoder, KBinsDiscretizer, KNNImputer, and IterativeImputer, are not evaluated for downstream utility or approximation quality.

The communication-cost measurements are useful, but they do not show that these methods produce accurate or useful preprocessing in realistic federated learning pipelines. For a paper that advertises a broad preprocessing framework, I would expect empirical validation of more than encoding and scaling.

## Missing experimental details and parameter studies

There are also reproducibility gaps. The paper describes IID and non-IID client partitioning, including a Dirichlet label-skew parameter, but I do not see the number of clients used in the main accuracy experiments. This is important because the number of clients affects FedAvg behavior, client heterogeneity, and the severity of local preprocessing inconsistency.

I would also expect experiments varying the number of clients and the degree of non-IIDness. Without these studies, it is difficult to know whether the observed local-vs-federated preprocessing gap is robust or specific to one partitioning setting.

## Dataset scope

The experiments focus on tabular datasets. This is suitable for many of the proposed preprocessing methods (especially in vertical FL). However, the title and framing should make this scope clear. The non-IID preprocessing problem is also important in other domains, such as physiological or biomedical signals, so the paper should either justify the exclusive focus on tabular data or make the tabular-data scope explicit in the title and claims.

## About privacy

The paper should be much more forward with the privacy issue. As now it is well-known that FL does not automatically provide privacy, it only avoids direct raw-data centralization. In this paper, some of the shared quantities, such as X^T X or XX^T, may leak substantial information about the underlying data. In some settings, sharing them could be more privacy-sensitive than sharing standard FL model updates.

This issue is particularly important in the vertical FL setting. It assumes that one party sends Y to the server. This is a very strong assumption. In many vertical FL applications, the label-holding party owns the most sensitive and commercially valuable information. Sending Y to the server may undermine both the privacy and the competitive motivation for using vertical FL.


## Comparison to competitors

The comparison to existing systems is quite short, and based on my checks it appears incomplete. I was surprised by the paper's framing that federated preprocessing has not really been addressed before. There are several systems and works that already handle aspects of federated preprocessing or federated data preparation.

Could the authors double-check the claims in Table 6? I found some methods that appear to be marked as unsupported even though related functionality exists in the cited competitors (although double-checking is needed to make sure that they are indeed in FL).

For example:
- FATE has OneHotEncoder and HomoOneHotEncoder:
https://github.com/FederatedAI/FATE/blob/v1.11.0/python/federatedml/feature/one_hot_encoder.py
- FATE has missing-value filling through DataTransform, which is at least partially related to SimpleImputer:
https://fate.readthedocs.io/en/v1.11.0/federatedml_component/data_transform/#federatedml.param.data_transform_param.DataTransformParam
- SecretFlow lists LabelEncoder and MinMaxScaler in its preprocessing API:
https://secret-flow.antgroup.com/docs/secretflow/zh_CN/source/secretflow.html

**Requested Changes:**

## Important changes

- Clarify the main contribution.

The authors should state more explicitly whether the contribution is primarily a software library, a general framework, new federated algorithms for specific preprocessors, or an empirical study of preprocessing in FL. The current paper mixes these aspects, but does not clearly separate what is new from SOTA.

- Revise Table 6 where needed.

-  Expand the empirical evaluation or narrow the claims.

If the paper claims to provide a comprehensive federated preprocessing suite, the experiments should evaluate more than OrdinalEncoder and StandardScaler. At minimum, the paper should include representative experiments for one sketch-based method, one discretization or transformation method, one imputation method, and one high-cardinality categorical method.


-  Add missing-data experiments.

Since imputation is a major advertised component, the paper should evaluate SimpleImputer, KNNImputer, and/or IterativeImputer. These experiments should compare no imputation, local imputation, and federated imputation under IID and non-IID partitions.

- Report the number of clients.

The paper should explicitly state the number of clients used in all experiments. This is necessary for reproducibility and for interpreting the local-vs-federated preprocessing comparison.


- Clarify the scope of the paper in terms of datasets.

Since the experiments and most methods are focused on tabular data, the title and claims should reflect this scope. If the authors intend the method to apply beyond tabular data, this should be supported by experiments.


## Changes that would strengthen the paper

- Add parameter studies.

The paper should vary at least the number of clients and the non-IID Dirichlet parameter. These are central parameters in FL and can substantially affect the conclusions.


- Be more forward with privacy risk.

The paper should state that many methods proposed exacerbate the privacy risk of FL.

- Reconsider the vertical FL setting.

The assumption that one party sends Y to the server should be justified or replaced by a privacy-preserving alternative. In many vertical FL applications, the label vector is highly sensitive and should not be revealed to the server.

---

> ### Author Response · Authors · 2026-07-09
>
> We thank the reviewer for this feedback.
>
> > Clarify the main contribution.
>
> We revised the introduction to better present our contributions. FedPS is presented primarily as a preprocessing framework and software library that systematically maps preprocessing operators to federated computation primitives. Existing federated methods (e.g., federated k-Means and k-NN) are incorporated where appropriate, while new federated algorithms are developed when necessary, most notably our federated Bayesian linear regression formulation used by IterativeImputer.
>
> > Revise Table 6 where needed.
>
> We’ve double-checked and updated Table 6 to ensure that the claims are accurate. Meanwhile, we have added the latest version of these two packages that we currently compare.
>
> > Expand the empirical evaluation or narrow the claims.
>
> We have added experiments that measure the approximation quality in Table 6. We have paid particular attention to estimations of statistics that use sketch-based methods. The results confirm that the error of transformed data between federated preprocessing and centralized processing is very small.
>
> > Add missing-data experiments.
>
> In order to keep the paper focused and within the length guidelines for TMLR, we prefer not to add these experiments.  We think that comparing the effect of no imputation with local and federated imputation goes beyond the intended scope of our paper. The general value of imputation is already demonstrated in prior work, and our other experiments already provide us with sufficient evidence that federating pre-processing is preferable to local approaches.  In other words, our goal is not to compare the utility of every imputation method in the federated setting. Rather, our goal here is to show how we can make these (and other) methods available in the federated setting.
>
> > Report the number of clients.
>
> We have reported the number of clients in section 5. We set the number of clients to 30 and 10. The 10 clients setting is the new experiments we added.
>
> > Clarify the scope of the paper in terms of datasets.
>
> We have updated the title and content that the paper is aimed at structured data, e.g., tabular datasets.
>
> > Add parameter studies (vary the number of clients and non-IID Dirichlet parameter)
>
> We have added new experiments by varying the number of clients, see Table 4 and Appendix D. The results are consistent with our findings: federated preprocessing consistently improves model accuracy, while inconsistent local preprocessing can lead to performance degradation under non-IID data distributions.
>
> We did not find that varying the Dirichlet parameter provided any deeper insights.   Since our experiments already include both IID partitioning and a commonly used label skew setting with Dirichlet parameter $\alpha=0.5$ to provide a representative heterogeneous data distribution, we have not added any further plots on this aspect. However, we have added new experimental results showing the impact of feature skew setting.
>
> > Be more forward with privacy risk.
>
> We have added the discussion at the end of section 3.1 as well as in section 6. Our privacy model in this work is data minimization which aims to minimize communication costs. While this does not provide formal privacy guarantees, the methods we present can be implemented using cryptographic primitives for statistics aggregation. In order to keep the paper focused, we leave further study of private federated preprocessing for future work.
>
> > Reconsider the vertical FL setting.
>
> We have added a discussion at the end of section 4 to elaborate on the potential privacy risk.  Our analysis is that this can be mitigated by adopting known cryptographic techniques to protect the computation.